# Ground-based Investigation of HOx and Ozone Chemistry in Biomass Burning Plumes in Rural Idaho

Andrew J. Lindsay[1], Daniel C. Anderson[1,*,**], Rebecca A. Wernis[2,3], Yutong Liang[3], Allen H. Goldstein[2,3], Scott C. Herndon[4], Joseph R. Roscioli[4], Christoph Dyroff[4], Ed C. Fortner[4], Philip L. Croteau[4], Francesca Majluf[4], Jordan E. Krechmer[4], Tara I. Yacovitch[4], Walter B. Knighton[5], Ezra C. Wood[1]

[1]Department of Chemistry, Drexel University, Philadelphia, PA, USA
[2]Department of Civil and Environmental Engineering, University of California Berkeley, Berkeley, CA, USA
[3]Department of Environmental Science, Policy and Management, University of California Berkeley, Berkeley, CA, USA
[4]Aerodyne Research Inc., Billerica, MA, USA
[5]Department of Chemistry and Biochemistry, Montana State University, Bozeman, MT, USA

*Now at: University of Maryland, Baltimore County, Baltimore, MD, USA
**Now at: NASA Goddard Space Flight Center, Greenbelt, MD, USA

*Correspondence to*: Ezra C. Wood (ew456@drexel.edu)

**Abstract.** Ozone ($O_3$), a potent greenhouse gas that is detrimental to human health, is typically found in elevated concentrations within biomass burning (BB) smoke plumes. The radical species OH, $HO_2$, and $RO_2$ (known collectively as ROx) have central roles in the formation of secondary pollutants including $O_3$ but are poorly characterized for BB plumes. We present measurements of total peroxy radical concentrations ($[XO_2] \equiv [HO_2] + [RO_2]$) and additional trace-gas and particulate matter measurements from McCall, Idaho during August 2018. There were five distinct periods in which BB smoke impacted this site. During BB events, $O_3$ concentrations were enhanced as evidenced by ozone enhancement ratios ($\Delta O_3 / \Delta CO$) that ranged up to 0.06 ppbv ppbv$^{-1}$. $[XO_2]$ was similarly elevated during some BB events. Overall, quantified instantaneous ozone production rates ($P(O_3)$) were minimally impacted by the presence of smoke as NOx enhancements were minimal. Measured $XO_2$ concentrations were compared to zero-dimensional box modeling results to evaluate the effectiveness of the Master Chemical Mechanism (MCM) and GEOS-Chem mechanisms overall and during periods of BB influence. The models consistently over-estimated $XO_2$ with the base MCM and GEOS-Chem $XO_2$ predictions high by an average of 28% and 20%, respectively. One period of BB influence had distinct measured enhancements of 15 pptv $XO_2$ that were not reflected in the model output, likely due to the presence of unmeasured HOx sources. To our knowledge, this is the first BB study featuring peroxy radical measurements.

## 1 Introduction

Unprecedented wildfire activity has been observed in recent years. For example, Brazil's Amazon rainforest wildfires in 2019 (Cardil et al., 2020) and Australia's bush fires in 2019-2020 (Yu et al., 2020) were both marked by historically high amounts of land burned. The United States has also observed several staggering wildfire seasons characterized by hundreds of fatalities and tens of thousands of homes destroyed. For example, the 2018 season included California's deadliest fire ever (the Camp Fire) responsible for 85 deaths (CAL-FIRE, 2021), burned a total of 3.5 Mha of land, and elicited government spending of 3.1 billion dollars for fire suppression (NIFC, 2019). Due to climate change, the impacts associated with wildfire are projected to worsen. Many regions, including much of the Western US, are expected to be warmer and drier causing increases in fuel aridity. These conditions are conducive to increases in land area burned, extended fire seasons, and frequent extreme fires (Spracklen et al.,

2009;Yue et al., 2014;Yue et al., 2013;Goss et al., 2020). As a result, the amount of land area burned in the Western US is expected to increase by 24 – 124% by the mid-21$^{st}$ century (Yue et al., 2014;Yue et al., 2013).

Wildfire smoke degrades air quality both locally and far downwind. Wildfire smoke is associated with respiratory and cardiovascular health risks (Liu et al., 2015;Reid et al., 2016;DeFlorio-Barker et al., 2019;Stowell et al., 2019). These health risks will result in modest increases of respiratory illness hospitalizations as wildfire smoke will more frequently impact densely populated areas (Liu et al., 2016a). Smoke from biomass burning (BB), a term that includes wildfires, changes in composition as it is transported downwind. The initial composition is based on direct BB emissions: particulate matter (PM), NOx, and organic compounds spanning a wide range of volatilities. These emissions then transform through photochemical reactions that form secondary pollutants such as ozone ($O_3$) and secondary organic aerosol (SOA). The formation of $O_3$ within BB smoke is of concern because $O_3$ poses additional health concerns and is a prominent greenhouse gas. While the large quantities of PM within BB smoke are primarily responsible for BB smoke health concerns (Liu et al., 2015;DeFlorio-Barker et al., 2019;Reid et al., 2016), health risks from $O_3$ exposure include lung irritation, decreases in lung and cardiac function, higher susceptibility to respiratory infection, and early mortality (Bell et al., 2006;Park et al., 2004;Jerrett et al., 2009;Turner et al., 2016;Silva et al., 2013). Tropospheric ozone is the third most important anthropogenic greenhouse gas after $CO_2$ and $CH_4$. Increases in tropospheric ozone between the pre-industrial era and the present, from 1750 to 2011, have accounted for 0.40 W m$^{-2}$ of radiative forcing. For perspective, $CH_4$ and $CO_2$ increases are responsible for 0.48 W m$^{-2}$ and 1.82 W m$^{-2}$ of radiative forcing, respectively (Stocker et al., 2013).

Mitigating $O_3$ pollution will be increasingly challenging as background $O_3$ concentrations are increasing in some regions. Long-term aircraft observations across the Northern Hemisphere indicate average increases of tropospheric $O_3$ of 5% per decade (Gaudel et al., 2020). More specifically, Western US $O_3$ concentrations have been increasing 0.41 ppbv yr$^{-1}$ despite declining concentrations in the Eastern US (Cooper et al., 2012). As biomass burning accounts for an estimated 3.5% of global tropospheric $O_3$ formation (Jaffe and Wigder, 2012), this source of $O_3$ will be more important with projected wildfire activity (Jacob and Winner, 2009;Yue et al., 2015). Cities in the western US, which often have existing $O_3$ pollution issues, have their air quality exacerbated by BB smoke. For example, a study focused on eight predominantly western US cities found wildfire smoke to correlate with 19% of exceedances to the previous 75 ppbv $O_3$ NAAQS (National Ambient Air Quality Standard) standard despite smoke only being present 4.1% of the total days studied between May and the end of September (Gong et al., 2017). This NAAQS $O_3$ standard is in terms of maximum 8-hour daily concentration and has been set to a more stringent standard of 70 ppbv since 2015.

The correlation between $O_3$ concentrations and BB smoke is not completely straightforward. Depleted $O_3$ concentrations can be found in freshly emitted smoke plumes due to NO emissions reacting with background $O_3$. Aged plumes have also been observed with depleted $O_3$. This has been attributed to meteorological conditions (Wentworth et al., 2018), NOx sequestration via peroxy acetyl nitrate (PAN) formation (Alvarado et al., 2010), and significant PM emissions that attenuate sunlight and limit photochemistry (Xu et al., 2021). Ozone depletion is further exemplified by the wide variation in reported ozone enhancement ratios ($\Delta O_3/\Delta CO$). While most reported ratios are positive indicating net $O_3$ formation, negative $\Delta O_3/\Delta CO$ values have also been reported and indicate depletion. $\Delta O_3/\Delta CO$ ratios in temperate and boreal forest (those typical of the United States) are on average 0.018, 0.15, and 0.22 ppbv ppbv$^{-1}$ for smoke plumes aged < 2 days, 2-5 days, and > 5 days, respectively (Jaffe and Wigder, 2012). The wide variation in net $O_3$ production within BB smoke is related to several factors including fire dynamics, the extent of emissions, and meteorological conditions. Each of these factors influence the underlying ROx ("ROx" = OH, $HO_2$, and $RO_2$) chemistry that controls oxidation processes and secondary pollutant formation.

ROx chemistry has rarely been studied within BB plumes. The peroxy radicals $HO_2$ and $RO_2$ (R = organic group) oxidize nitric oxide (NO) to form $NO_2$ (R1 and R2). During the day, the resulting $NO_2$ is converted to $O_3$ by photolysis (R3 and R4):

$$HO_2 + NO \rightarrow OH + NO_2 \qquad (R1)$$

$$RO_2 + NO \rightarrow RO + NO_2 \qquad (R2)$$

$$NO_2 + h\nu \rightarrow NO + O(^3P) \qquad (R3)$$

$$O(^3P) + O_2 + M \rightarrow O_3 + M \qquad (R4)$$


Within these smoke plumes, the concentration and composition of ROx species depends on fire emissions, photochemical conditions, and smoke age. BB emissions include direct HOx precursors, but individual emission factors and ratios are highly variable between fires. Fire dynamics alone have a significant effect on emissions. Flaming conditions have greater combustion efficiencies and are characterized by smaller emission ratios of VOCs and PM, while reactive nitrogen emissions are dominated

by HONO, NO, and $NO_2$ (Burling et al., 2010;Roberts et al., 2020). Smoldering fire conditions have greater VOC emission ratios, PM emissions, but lower NOx emissions (Yokelson et al., 1996). Different fuel sources, such as unique tree species, shrubs, grasses, and crops, have unique emission profiles (Koss et al., 2018). Direct HOx precursor emissions of formaldehyde (HCHO), acetaldehyde ($CH_3CHO$), and nitrous acid (HONO) are greater than those from more typical urban combustion sources. HONO within BB plumes is short lived with a chemical lifetime ranging 10 to 30 minutes due to photolysis. This rapid photolysis yields

OH and NO and has caused HONO to act as the dominant ROx source in some freshly emitted smoke plumes (Peng et al., 2020;Yokelson et al., 2009). Ozone formation in young BB plumes is, in almost all cases, initially NOx-saturated (VOC-limited) but transitions to being NOx-limited as the NOx is photochemically processed to nitric acid and organic nitrates (Xu et al., 2021;Alvarado et al., 2015;Müller et al., 2016;Folkins et al., 1997). In particular, $NO_2$ can be efficiently sequestered in the form of peroxyacetyl nitrate (PAN) due to the importance of the acetaldehyde emissions and the relative importance of the $CH_3C(O)OO$

radical (Peng et al., 2021). The subsequent thermal decomposition of PAN into air masses in which ozone production is NOx-limited can potentially lead to sustained $O_3$ formation far downwind of a fire. BB emissions also include unique VOCs that are typically unaccounted for by chemical mechanisms employed by models. For instance, the importance of furanoids for model predictions of secondary pollution formation has only recently been studied (Müller et al., 2016;Coggon et al., 2019;Decker et al., 2019;Salvador et al., 2021;Robinson et al., 2021).

For the best understanding of the composition and concentration of radicals, direct measurements and improved models are necessary. Measurements of any ROx compounds within BB plumes are rare. Only one study has reported direct measurements of OH in a BB smoke plume, which found that freshly emitted BB plumes (22-43 minutes aged) had OH concentrations five times greater than that of background air (Yokelson et al., 2009). Our understanding of ROx chemistry in BB smoke has historically relied on calculations and models (Mason et al., 2001;Hobbs et al., 2003;de Gouw et al., 2006;Akagi et al., 2012;Liu et al.,

2016b;Müller et al., 2016;Parrington et al., 2013). Elevated OH concentrations have been suggested for freshly emitted smoke (Hobbs et al., 2003;Akagi et al., 2012), while especially low OH concentrations were calculated for aged plumes of ~4 days in western Canada (de Gouw et al., 2006). There are few studies that have focused on peroxy radical chemistry within BB studies and it appears there have been no direct measurements in BB smoke. Model suggested peroxy radical concentrations have exhibited a wide range of values (Mason et al., 2001;Parrington et al., 2013;Liu et al., 2016b;Baylon et al., 2018), in some cases reaching

unrealistically high values ([$HO_2 + RO_2$] >> 200 pptv for wildfires in Nova Scotia, Canada as presented in Parrington et al. (2013)).

This manuscript focuses on smoke observations collected in McCall, Idaho in the Pacific Northwest – a region particularly prone to wildfire – as part of the joint NCAR WE-CAN (Western Wildfire Experiment for Cloud Chemistry, Aerosol Absorption, and Nitrogen) and NOAA FIREX (Fire Influence on Regional to Global Environments Experiment) study. Increases in wildfire activity are anticipated for parts of this region, including much of Idaho, because of climate change (Halofsky et al., 2020).

Presented are possibly the first measurements of total peroxy radicals in biomass burning plumes, enabling a unique investigation

into the impacts of biomass burning on photochemistry and ozone production and the accuracy of commonly used atmospheric chemistry models.

## 2 Methods

### 2.1 Campaign Description

Measurements were collected in McCall, Idaho during the WE-CAN/FIREX 2018 campaign. McCall (elevation ~1.5 km) is a rural town in Valley County, Idaho approximately 160 km north of Boise, Idaho (Fig. 1) within the West Mountains of Idaho. While much of the local area is used for cattle grazing, the town attracts tourists year-round for outdoor recreation due to the presence of mountains, surrounding forests, and a large lake.

The McCall field site included two mobile laboratories and one building. Each were outfitted with instrumentation for
gas- and particle-phase measurements. The largest mobile lab was the Aerodyne Mobile Laboratory (AML) (Herndon et al., 2005). The AML split its time between stationary sampling in McCall and mobile measurements in other parts of Idaho and surrounding states. Unlike the AML, the second mobile lab, known as the Miniature Aerodyne Mobile Lab (minAML), was permanently stationed at the McCall site. Our analyses focus on the date ranges 16-18 August and 21-24 August when the AML was stationed at the McCall site.

**2.2 ECHAMP**

ECHAMP (Ethane CHemical AMPlifier) is a chemical amplification-based instrument that was used to measure total peroxy radical concentrations ($[HO_2] + [RO_2]$, or simply $[XO_2]$). ECHAMP was stationed within the minAML and sampled on a 2-minute time base. The sampling and calibration methods have been described in detail elsewhere (Anderson et al., 2019;Wood et al., 2017) and only a brief summary including details specific to the McCall deployment is described here.

ECHAMP 'amplifies' each sampled $XO_2$ molecule into a greater number of $NO_2$ molecules. The enhancement in $NO_2$ concentration from 'amplification' is then divided by an amplification factor to determine $[XO_2]$. Amplification is achieved by mixing sampled air with elevated concentrations of NO and $C_2H_6$ to take advantage of a radical propagation scheme (Reactions 1-2 and 5-8). Since these reactions can proceed multiple times, each $XO_2$ produces up to 20 $NO_2$ molecules. There are two sampling channels: in the amplification channel there is an immediate addition of $C_2H_6$ and NO, whereas in the background channel the
$C_2H_6$ addition is delayed to avoid Reaction 5, and all peroxy radicals are converted into HONO (Wood et al., 2017). The $NO_2$ within the two channels are transported to and measured by respective Cavity Attenuated Phase Shift (CAPS) $NO_2$ monitors (Kebabian et al., 2005).

$$OH + C_2H_6 \rightarrow H_2O + C_2H_5 \tag{R5}$$
$$C_2H_5 + O_2 + M \rightarrow C_2H_5O_2 + M \tag{R6}$$
$$C_2H_5O_2 + NO \rightarrow C_2H_5O + NO_2 \tag{R7}$$
$$C_2H_5O + O_2 \rightarrow CH_3CHO + HO_2 \tag{R8}$$

Ambient air is first sampled at 2.5 SLPM into a ¼" PFA tee and immediately diluted with 0.8 SLPM $O_2$. The sampled air then
flows through 10 cm of 0.635 cm OD PFA tubing protruding out of an inlet box.  This ECHAMP inlet box is a weatherproof container with dimensions of 39 cm × 44 cm × 16 cm and was mounted 3 m above ground level on scaffolding. The sampled air

then entered a glass cross that was internally coated with halocarbon wax. Connected orthogonally from the sampling lines were two 15.2 cm long (0.635 cm OD) reaction chambers. The reaction chambers each subsampled at a flow rate of 0.83 lpm. The remaining flow rate of 1.64 SLPM traveled by a Vaisala HMP60 probe that measured temperature and humidity. The reaction chambers included additions of NO, $C_2H_6$, and $N_2$. In amplification mode, 20 sccm of 39.3 ppmv NO in $N_2$ and 50 sccm of 25% $C_2H_6$ in $N_2$ were added at the beginning of the reaction chamber, while 50 sccm of $N_2$ is added in a downstream position. In the background mode, the locations for the $N_2$ and $C_2H_6$ additions are reversed. The final concentrations of NO and $C_2H_6$ after reaction chambers were 0.827 ppmv and 1.32%, respectively. These reaction chambers alternate in one-minute intervals between background and amplification modes leading to the overall two-minute sampling time. The resulting mixtures from both channels were transported in approximately 23 m of their respective tubing to the minAML that housed both CAPS monitors.

Dilution of sampled air with $O_2$ is a new addition to ECHAMP and is similar in some ways to its use by the perCIMS method (Hornbrook et al., 2011). The elevated $O_2$ concentration (40%) increases the ratio of the rate of the propagation reaction R8 ($C_2H_5O$ + $O_2 \rightarrow CH_3CHO + HO_2$) to the rate of ethyl nitrite formation ($C_2H_5O + NO + M \rightarrow C_2H_5ONO + M$) which is a termination reaction. Dilution also dries the sampled air leading to lower and less variable relative humidity. This is beneficial as the amplification factor decreases with increasing RH (Anderson et al., 2019;Wood et al., 2017). The humidity-dependence of the amplification factor is due to humidity dependent reactions of $HO_2$ and the $HO_2$-$H_2O$ adduct reacting with NO to form $HNO_3$ (Reichert et al., 2003;Butkovskaya et al., 2009) and $HO_2$ wall losses (Mihele and Hastie, 1998;Reichert et al., 2003).

ECHAMP was calibrated to the methyl peroxy radical $CH_3O_2$ over a range of relative humidities (RH) using the $CH_3I$ photolysis method as described in Anderson et al. (2019) six times at McCall. The calibrant was prepared by mixing humidified ZA with trace amounts of $CH_3I$ from a permeation source. The resulting mixture flows to a quartz tube where $CH_3I$ is photolyzed at 254 nm by an $O_3$-free Hg lamp. The resulting $CH_3O_2$ calibrant is then added in excess flow to the ECHAMP inlet. To quantify the $CH_3O_2$ produced, both reaction chambers were initially operated in background mode and the $CH_3O_2$ source modulated on and off by alternating the flow between UV photolysis cell and a by-pass chamber. An improvement over the prior version of this calibration method was the elimination of dead-volume in the chamber. Further details regarding this calibration technique including recent improvements are provided in the supplement. Calibrations were also conducted using the $H_2O$ photolysis method (Anderson et al., 2019). Unfortunately, inconsistent results were obtained, and at the end of the project we discovered that the quartz photolysis cell was broken. Therefore, calibrations using the $H_2O$ photolysis method were disregarded and only the $CH_3I$ calibrations were used. Based on the uncertainties in the individual calibration points, the variability among individual calibration points, and uncertainties regarding sampling losses we ascribe an uncertainty of 34% ($2\sigma$) to the measurements. See SI Sect. S1 for more information.

## 2.3 Additional Measurements

Onboard the AML, Quantum Cascade Tunable Infrared Laser Direct Absorption Spectrometers (QC-TILDAS, Aerodyne Research Inc.) (McManus et al., 2015) were used to measure 1) NO, $NO_2$; 2) CO, $N_2O$, $H_2O$; 3) HCHO, HCOOH; 4) $CH_4$, $C_2H_6$; and 5) HCN. Ozone was measured by a 2B-Tech UV absorption instrument. While particulate matter and VOCs can positively interfere with photometric $O_3$ measurements (Huntzicker and Johnson, 1979;Long et al., 2021), comparison to a separate Ox measurement revealed minimal interferences in the 2B-Tech $O_3$ observations (see Sect. S2 of SI). VOC measurements were made by an ARI Vocus (proton-transfer-reaction high-resolution time-of-flight (PTR-HR-ToF) mass spectrometer) (Krechmer et al., 2018). Measured VOCs include isoprene, acetaldehyde, acetone, acetic acid, benzene, toluene, C2-benzenes, C3-benzenes, methanol, total monoterpenes, and the sum of methyl vinyl ketone (MVK) and methacrolein (MACR). BB-related VOCs measured

with the Vocus include furan, methyl furan, furfural, the sum of methyl furfural and catechol, and guaiacol. Chemically-resolved measurements of particulate matter mass concentrations were made by an ARI Soot Particle Aerosol Mass Spectrometer (SP-AMS) (Canagaratna et al., 2007). Additional particle-phase measurements were made with an ARI Aerosol Chemical Speciation Monitor (ACSM) permanently stationed at the McCall site building (Ng et al., 2011).

Onboard the minAML, $NO_2$ was measured with a CAPS monitor and VOCs with the Berkeley Comprehensive Thermal Desorption Aerosol Gas Chromatograph (cTAG) (Wernis et al., 2021). cTAG measures concentrations of VOCs, intermediate volatility organic compounds and semi-volatile organic compounds spanning an alkane-equivalent volatility range from $C_5$ to $C_{30}$ every hour via pre-concentration followed by thermal desorption and gas chromatography-time-of-flight mass spectrometry (GC-TOFMS). This manuscript uses isoprene, speciated monoterpenes, 2-methyl-3-buten-2-ol (MBO), styrene, benzene and toluene measurements taken by cTAG.

Meteorological measurements were made both on the AML and permanently at the McCall site. Temperature, wind speed, and wind direction were collected by a 3-D R.M. Young (Model 81000RE) sonic anemometer stationed permanently at the McCall site at a height of 10 m. Additional wind was measured with a 2D R.M. Young (Model 81000RE) sonic anemometer mounted to the AML rooftop and corrected for speed and truck orientation with data from a Hemisphere GPS compass (model Vector V103). Temperature, RH, and wind data are shared in the supporting information (Fig. S3). Daily maximum temperatures ranged 22 °C and 28 °C while minimum temperatures ranged 4 °C to 13 °C. Solar irradiance was measured by a permanently stationed ARISense air quality sensor system (Cross et al., 2017). This was used to derive photolysis frequencies of interest, such as $J_{NO_2}$, by scaling measured irradiance to outputs from the National Center for Atmospheric Research (NCAR) Tropospheric Ultraviolet and Visible (TUV) radiation model. This process for deriving photolysis frequencies is described in greater detail in the supplement.

For most calculations and chemistry analyses, measured concentrations of other compounds were synchronized to the 2-minute ECHAMP time scale. This was achieved by averaging greater frequency measurements and linearly interpolating lower frequency measurements. Most measurements fall in the former category and were measured at a 1 Hz sampling rate. VOCs from cTAG were obtained at an hourly rate and were therefore linearly interpolated.

## 2.4 Smoke Events

Time periods impacted by smoke were identified with observations of the biomass burning tracers HCN, $CH_3CN$, organic aerosol (OA), and CO. HCN was used as the primary tracer for BB smoke. Nitriles are commonly used as tracers for BB, and HCN is particularly useful in the absence of nearby vehicle sources. Emissions of HCN are essentially inert within BB plumes with an atmospheric lifetime of 2 to 4 months (Li et al., 2000). HCN emission ratios are dependent on the biomass burning fuel type (Koss et al., 2018;Coggon et al., 2016) and can vary with fire dynamics (Roberts et al., 2020). Figure 2 presents McCall site observations and indicates smoke-impacted time periods when ECHAMP was actively sampling and the AML was present. The lowest concentrations of HCN, $CH_3CN$, CO, and OA were all observed on 24 August, suggesting the air sampled up until then was always somewhat affected by BB emissions. We experienced 5 distinct smoke-impacted periods evident by clear enhancements of HCN along with organic PM and CO. The periods of greatest smoke influence (red shaded regions in Fig. 2) were identified by sustained periods of HCN concentrations greater than 1 ppbv. General smoke presence (tan shaded regions in Fig. 2) was identified before and after each significant smoke period when background smoke tracer concentrations remained elevated compared to stable background air. There were distinct smoke periods in the early evenings of 16 August and 17 August. These occurred from 18:48 MDT to 21:55 MDT on 16 Aug and 15:27 MDT to 18:15 MDT on 17 Aug. While appreciable enhancements in CO were observed during both of these periods, a significant OA enhancement of ~20 µg m$^{-3}$ was only observed during 17 August. For the period 21

to 24 August, smoke events occurred at earlier times. Periods of significant smoke influence on 22 August, 23 August, and 24 August were from 12:55 MDT to 16:34 MDT, 12:26 MDT to 17:07 MDT, and 7:42 MDT to 15:00 MDT, respectively. The 22 August smoke event had minimal enhancement of OA. The smoke event on 23 August was quite distinct with the greatest HCN concentrations observed for the entire campaign. After this 23 August significant smoke period highlighted in Fig. 2, concentrations of HCN, CO, and OA decreased but remained at levels above background concentrations until the 24 August event. Following the 24 August significant smoke period, CO and OA decreased to their lowest observed concentrations indicating that we were sampling an air mass with minimal smoke influence.

We identify the sources of the observed smoke by pairing NOAA Air Resources Laboratory HYbrid Single Particle Lagrangian Integrated Trajectory (HYSPLIT) (Stein et al., 2015) model back trajectories with satellite detected wildfire locations (Lindaas et al., 2017;Rogers et al., 2020). Panel A of Fig. 1 shows all wildfires detected for the date range we focus our analyses on. Three well documented fires are indicated: the Mesa Fire, the Rattlesnake Creek Fire, and the Rabbit Foot Fire. The Mesa Fire, located 38.3 km southwest of McCall, was responsible for burning 34,700 acres (~14,000 hectares) and started 26 July 2018 (FWAC, 2021b). The Rattlesnake Creek Fire started 23 July, 2018, burned over 8,000 acres (>3,300 hectares) (FWAC, 2021c). The Rabbit Foot Fire began on 2 August 2018, burned 36,000 acres (~15,000 hectares), that was not contained until November (FWAC, 2021a). Panel B of Fig. 1 shows 17 August as a representative smoke-impacted day. Shown are bihourly HYSPLIT 48 hour-long back trajectories initialized from the McCall site at 10 m above ground level using archived Global Data Assimilation System (GDAS) 1-degree meteorological data. Based on fire tracer observations (see Fig. 2), the 17 August significant smoke period started at ~15:30 MDT and persisted until 18:15 MDT. This is consistent with the 16:00 MDT and 18:00 MDT back trajectories that show air sampled during that period was transported past an active fire approximately 60 km southwest of the McCall, Idaho site. Though an uncontrolled portion of the Mesa Fire may have also contributed to these smoky conditions, the HYSPLIT trajectories would suggest earlier smoke influence possibly starting near 14:00 MDT. The smoke likely traveled between 3-5 hours to get to the McCall site from the unspecified active fire for the 17 August example shown. Figures showing HYSPLIT trajectories for other smoke influenced days are included in the SI (Sect. S4). The likely sources and smoke ages for other events are detailed here. The 16 August smoke event was likely sourced from the same unspecified wildfire ranging in age between 4-10 hours. The smoke on 22 August was likely 12-18 hours or longer and sourced from an unspecified fire located east between the McCall site and the Rabbit Foot Fire. The 23 August smoke was likely sourced from the southwest from another unspecified fire at an age of 12-18 hours, though the Mesa Fire may have contributed here. The smoky conditions of the morning of 24 August were likely from wildfires in central Oregon aged 18-30 hours until especially clean air was sourced from Northern Oregon beginning at approximately 12:00 MDT.

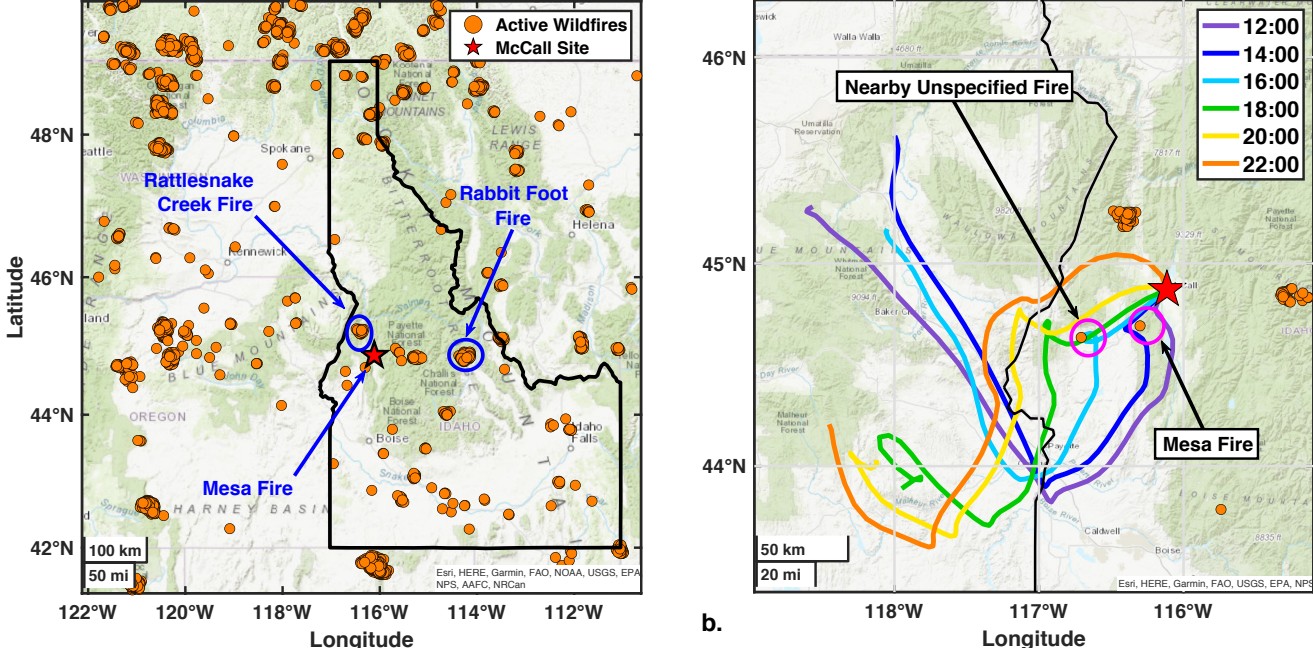

**Figure 1** The McCall, Idaho site is indicated by the red star and wildfire locations indicated by orange circles. Panel A shows detected wildfires between 15 August 2018 and 24 August 2018. Panel B includes bihourly HYSPLIT back trajectories for the representative smoke impacted day of 17 August. Shown here are active wildfires of 15 August through 17 August, as back trajectories end 24 hours back. Fire locations were retrieved using the NOAA Hazard Mapping System (HMS) (NOAA, 2021).

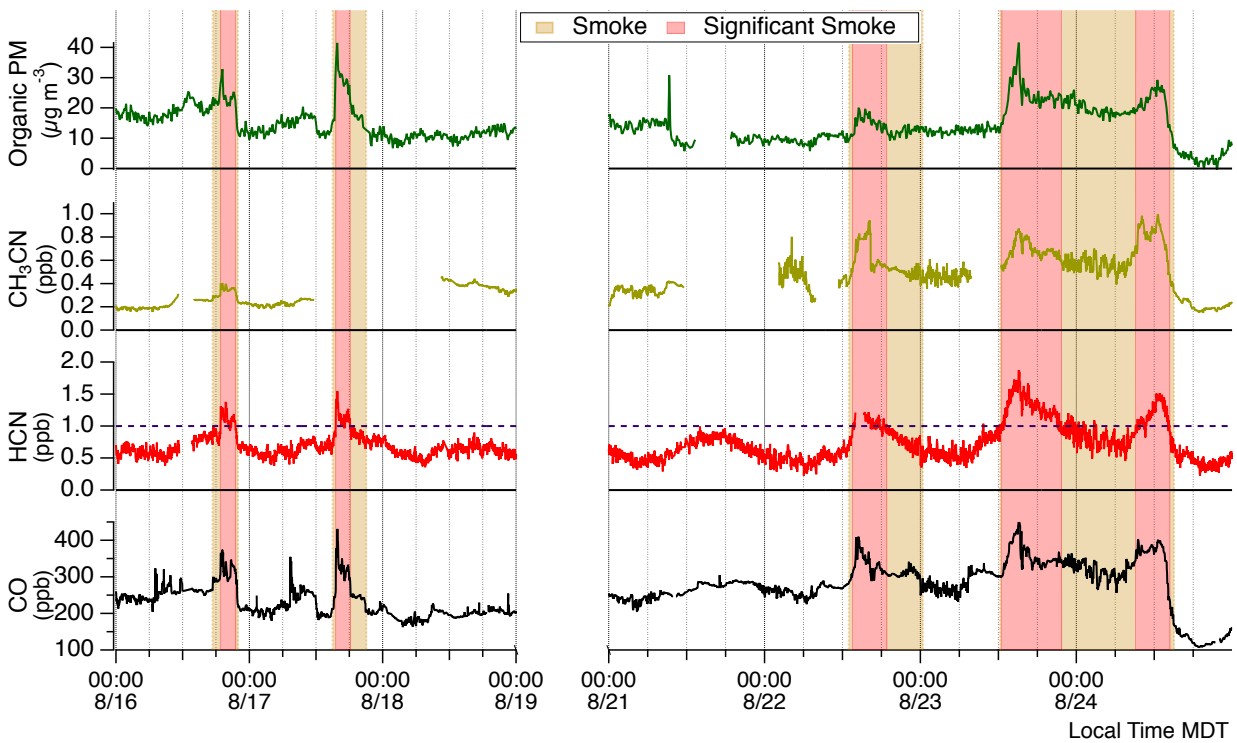

**Figure 2** Time series of smoke tracers. Tan shading represents smoke presence, while red shading signifies significant smoke periods. Significant smoke periods are defined by HCN concentrations greater than 1 ppbv. This smoke-defining concentration is indicated by the purple dashed line.

## 2.5 Calculations

Ozone enhancement ratios $\Delta O_3/\Delta CO$ from smoke influence were determined for the most distinct BB events of 16, 17, and 24 August. For the 17 and 24 August events, the ozone enhancement ratio was determined using the York bi-variate linear regression method (York et al., 2004) using a continuous section of $O_3$ and $CO$ data that includes 60 minutes of background air, a transitional smoke period (tan shaded regions in Fig. 2), and 60 minutes of significant smoke period data (red shaded regions in Fig. 2). Enhancements in $NO_2$ were typically under 0.2 ppbv and so the difference between considering $\Delta Ox$ ($Ox = O_3 + NO_2$) and $\Delta O_3$ was negligible. The linear regressions are included in the supplement (Fig. S8). $\Delta O_3/\Delta CO$ for the 16 August event was determined using Eq (1) with $O_3$ and $CO$ data collected during a stable period at the start of the significant smoke period and a stable background prior to smoke presence.

$$\Delta O_3/\Delta CO = ([O_3]_{Smoke} - [O_3]_{Background})/([CO]_{Smoke} - [CO]_{Background}) \quad (1)$$

This event had a temporary depletion in $O_3$ by ~20 ppbv for the start of smoke significance, then returned to near background levels of $O_3$. $\Delta O_3/\Delta CO$ values were not calculated for the remaining 22 August and 23 August smoke events. These events had less distinct $O_3$ enhancements and occurred at times that $O_3$ increased during non-smoky time periods.

The gross instantaneous $O_X$ ($[O_X] \equiv [O_3] + [NO_2]$) production rate ($P(O_X)$), often referred to as $P(O_3)$, is the rate at which NO is converted to $NO_2$ by reaction with $HO_2$ or $RO_2$ (Eq. 2). Since we measure the sum of $HO_2$ and $RO_2$, we calculate $P(Ox)$ with Eq. (3) using ECHAMP $XO_2$ and AML-based QC-TILDAS NO measurements. Noting that the differences between $k_{HO2+NO}$ and most $k_{RO2+NO}$ rate constants are small (Anderson et al., 2019), we use an effective rate constant ($k_{eff}$) equal to $k_{HO2+NO}$. $P(Ox)$ from box model results were calculated using Eq. (2) and the model $HO_2$ and speciated $RO_2$ concentrations.

$$P(Ox) = k_{HO2+NO}[HO_2][NO] + \sum k_{RO2(i)+NO}[RO_{2(i)}][NO] \quad (2)$$

$$P(Ox) = k_{eff}[XO_2][NO] \quad (3)$$

Instantaneous $ROx$ production rates $P(ROx)$ from measured compounds were calculated using Eq. (4). Each term represents a compound that undergoes photolysis to produce two $ROx$ radicals, with compound-specific photolysis rate constants (frequencies) indicated by the $J$ variables. Our calculated values for $P(ROx)$ are limited by the lack of measurements for HONO, which is the dominant $HOx$ source in freshly emitted BB smoke (Peng et al., 2020;Robinson et al., 2021). While HONO has a lifetime of ~20 minutes during daylight hours, dark plume conditions and possible photochemical formation on aerosol particles may lead to sustained HONO concentrations. Ozonolysis of measured alkenes had minimal contribution to daytime $P(ROx)$ values and was therefore omitted from this calculation but is included in model predictions.

$$P(ROx) = 2J_{O^1D}[O_3] \frac{k_{(O^1D+H_2O)}[H_2O]}{k_{(O^1D+H_2O)}[H_2O] + k_{(O^1D+N_2)}[N_2] + k_{(O^1D+O_2)}[O_2]} + 2J_{HCHO}[HCHO] + 2J_{CH_3CHO}[CH_3CHO] +$$
$$2J_{CH_3COCH_3}[CH_3COCH_3] \quad (4)$$

## 2.6 Zero-Dimensional Modeling

The Framework for 0-D Atmospheric Modeling (F0AM) box model (v3.2) (Wolfe et al., 2016) was used to evaluate ECHAMP $XO_2$ measurements and further investigate BB impacts on instantaneous chemistry. F0AM simulations were conducted for the ECHAMP 2-minute time basis for dates in which AML concomitant measurements were present. Modeling was conducted separately for the two date ranges of interest of 16 to 18 August and 21 to 24 August then combined for analysis. We primarily focus on results acquired by employing a subset of the Master Chemical Mechanism (MCM) version 3.3.1 (Saunders et al.,

2003;Jenkin et al., 2003;Jenkin et al., 2015) that included only the relevant chemical species in order to avoid unnecessary reactions and improve model time consumption. The model was constrained with all available measurements (see SI for full list), including concentrations of ozone, formaldehyde, acetaldehyde, acetone, isoprene, speciated monoterpenes, MVK, MACR, and MBO. While F0AM allows for total NOx to be constrained, we instead constrained NO and $NO_2$ individually. Model results obtained using total NOx constraints led to nearly identical daytime $XO_2$ predictions but with unrealistic nighttime $XO_2$ values. The base MCM mechanism, referred to as "MCM-base" from here on, was augmented for two additional F0AM simulations. First, the MCM-base was expanded by including additional chemistry for BB-related VOCs (referred to as "MCM-BBVOC") of furan, methyl furan, furfural, methyl furfural, and guaiacol by manually adding the relevant chemical reactions to the MCM as detailed by Coggon et al. (2019). Second, a mechanism referred to as "MCM-BBVOC-het" included heterogeneous chemistry for $HO_2$ loss on organic aerosols in addition to the previously detailed BB VOC chemistry. The heterogeneous loss rates are dependent on the predicted $[HO_2]$ values, organic aerosol surface area concentration, an uptake coefficient ($\gamma$), and mean molecular speed (Tang et al., 2014). Aerosol surface area concentrations were unmeasured and instead calculated from mass concentration measurements by applying a specific surface area. The default specific surface area was set to 4 $m^2$ $g^{-1}$. This setting falls slightly below the typical values measured for an urban environment of Tokyo, Japan (Hatoya et al., 2016). The default uptake coefficient was 0.20 as recommended by Jacob (2000). We explored the sensitivity of model results to both the specific surface area and uptake coefficient parameters by varying settings. We also share F0AM results acquired using the GEOS-Chem chemical mechanism. This version of the GEOS-Chem mechanism uses version 9-02 (Mao et al., 2013) with isoprene chemistry updates (Marais et al., 2016;Fisher et al., 2016;Travis et al., 2016;Kim et al., 2015). A small first-order dilution was implemented for all model experiments so that all compounds would have 24-hour lifetimes in order to prevent unreasonable accumulation of secondary species with background concentrations for all unmeasured compounds set to 0 ppbv (Wolfe et al., 2016). Concentrations of unmeasured species were also set to 0 ppbv for the model results presented in this manuscript. Minimal changes in model results were observed for additional simulations that included a "spin-up" period in order to determine initial concentrations of unmeasured compounds. As mentioned earlier, HONO is a particularly important ROx precursor in BB plumes. In addition to it not being measured during this study, the zero-dimensional models utilized cannot be expected to accurately predict HONO concentrations since a portion of the HONO in the sampled air masses was undoubtedly emitted directly by the smoke. Furthermore, there are no HONO formation processes in the chemical mechanisms besides its homogenous formation from the reaction of OH with NO (i.e., there are no heterogeneous formation mechanisms). A complete description of our model setup, including observational constraints and uncertainties, is provided in the SI (see Sect. S7).

## 3 Results and Discussion

### 3.1 Smoke Influence on ozone and its precursors

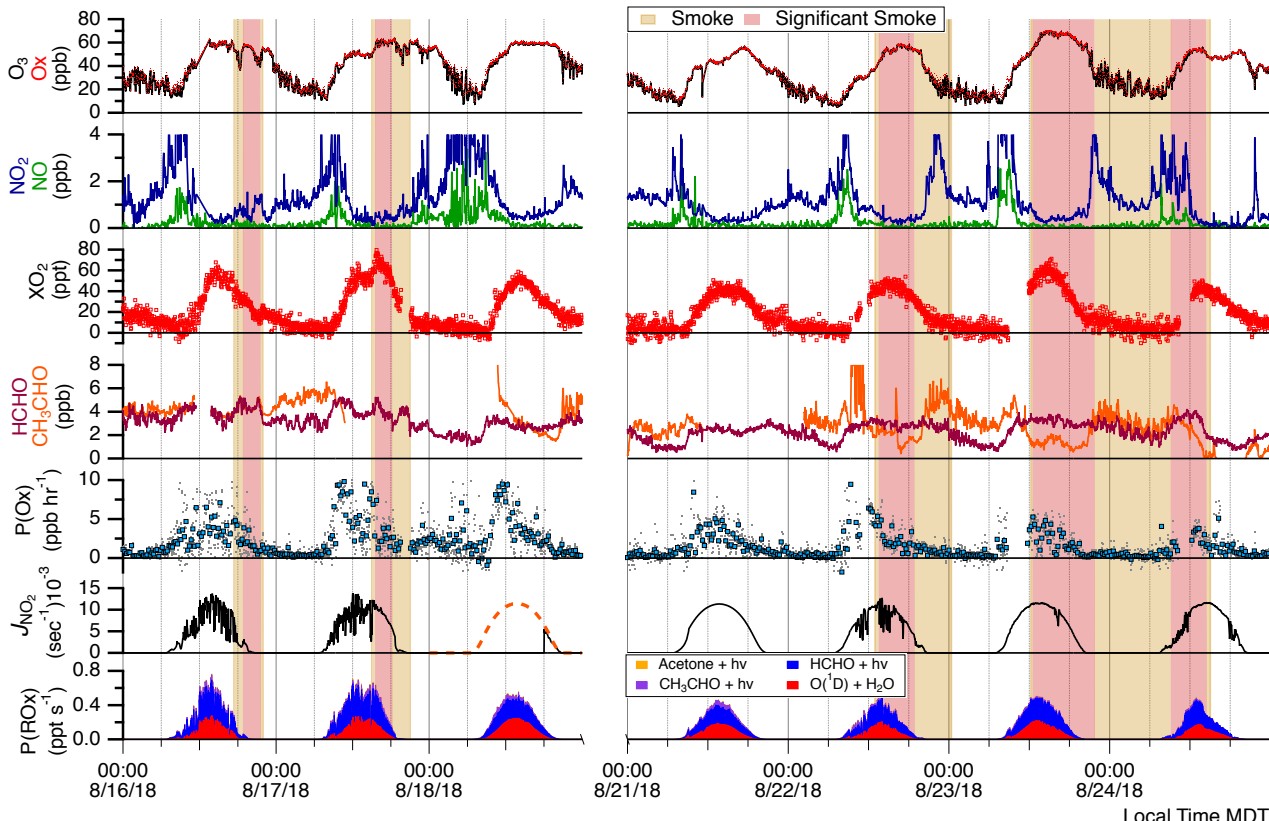

**Figure 3** Time-series of $O_3$, NO, $NO_2$, total peroxy radicals ($XO_2$), and aldehydes (HCHO and $CH_3CHO$). Modeled $J_{NO_2}$ frequencies and calculated parameters of P(Ox) and P(ROx) are also provided. The red shading indicates periods of smoke influence, while somewhat smoky periods are shaded tan. P(Ox) is shown in both 16-minute averages (blue circles) and 2-minute data (grey points). The dashed $J_{NO_2}$ trace is entirely simulated by NCAR TUV model whereas the rest of the data is derived from ARISense solar irradiance measurements.

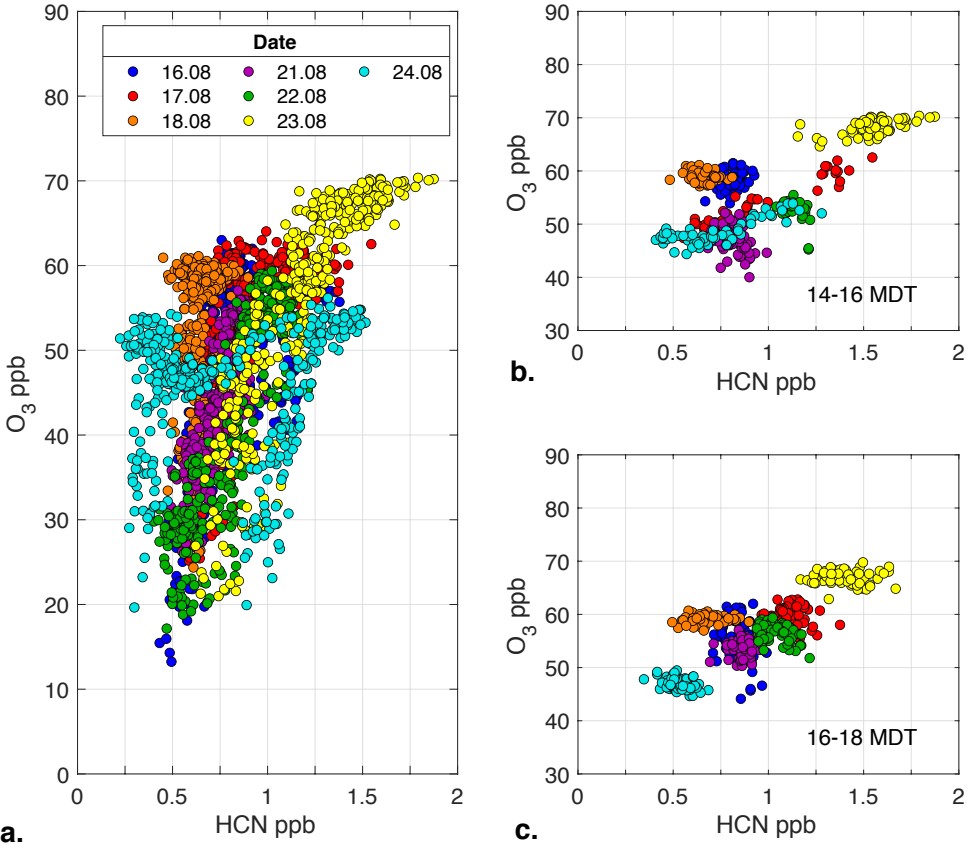

**Figure 4** Correlation between O$_3$ and smoke tracer HCN for all observations between 9:00 and 22:00 MDT (panel a), 14:00 and 16:00 MDT (panel b), and 16:00 and 18:00 MDT (panel c). Data points are colored by date collected.

Figure 3 shows site observations of O$_3$ and relevant measurements of NOx, total peroxy radicals, and aldehydes. Typically O$_3$ increased from ~20 ppbv overnight up to daily maximum concentrations between 50 and 60 ppbv, with the greatest O$_3$ concentrations of near 70 ppbv observed on 23 August. NOx concentrations were consistently low with typical daytime [NO] and [NO$_2$] values below 0.1 and 1.0 ppbv, respectively, suggesting O$_3$ production in the NOx limited regime (Sillman et al., 1990). Ozone production is further discussed in Sect. 3.2. XO$_2$ typically reached daily maximum concentrations between 40 and 60 pptv. Distinct increases in [O$_3$] and [XO$_2$] were observed during some BB events. The 17 August event had the clearest impact, with [O$_3$] increasing ~10 ppbv and [XO$_2$] increasing ~15 pptv over 45 minutes resulting in the maximum [XO$_2$] observed of near 70 pptv for the entire campaign. The 16 August and 24 August smoke events had distinct [O$_3$] increases but with less notable change in [XO$_2$]. For 24 August, smoke impacted the site throughout the morning and persisted into the afternoon. Ozone concentrations were ~53 ppbv for over an hour until smoky conditions dispersed. Ozone concentrations then decreased and remained near 46 ppbv, demonstrating a 7 ppbv elevation associated with the presence of smoke. No distinct impact on XO$_2$ was observed for this event. The smoke impact on 16 August was unique as [O$_3$] was depleted by ~10 ppbv upon smoke arrival, returned to near background concentrations, then was again depleted by ~10 ppbv as smoke exited. Enhancements in [NO$_2$] during these periods of depleted O$_3$ were small – only 0.5 and 1.0 ppbv – thus [Ox] was depleted as well. This variation in [O$_3$] and smoke tracers was likely due to different regions of this smoke plume impacting the site at different times. The edges of the plume may have been sampled at the beginning and end of the smoke period, and the plume center sampled during the period with the greatest O$_3$ concentrations. XO$_2$ responded similarly to O$_3$ and was depleted by ~10 pptv upon the arrival and departure of smoke. The 23

August event was particularly smoky with the greatest smoke tracer concentrations of HCN, CO, and organic PM as well as the highest concentrations of $O_3$ for the entire campaign.

For the entire campaign, there is a positive correlation between daytime $O_3$ (and Ox) concentrations and smoke tracer HCN (Fig. 4a) with the highest values for both observed on 23 August. Most periods of elevated HCN occurred during the times of day when $[O_3]$ was usually high even in the absence of smoke (afternoon or early evening), so the overall positive correlation between $O_3$ and HCN may be partially coincidental. The positive correlation remains, however, when the analysis is restricted to 2-hour periods of afternoon and early evening data to limit the time-of-day dependence (Fig. 4b and Fig. 4c). These more specific $O_3$-HCN comparisons remain impacted by day-to-day variability in $O_3$ from changes in background $O_3$ values, meteorology, and BB HCN and $O_3$ precursor emissions. Smoke age also plays a role in this correlation plot. Based on literature trends where $\Delta O_3/\Delta CO$ values increase with smoke age until an eventual plateau (Jaffe and Wigder, 2012;Baker et al., 2016;Xu et al., 2021), young smoke plumes are likely to have smaller $O_3$ enhancements relative to smoke tracers like HCN compared to aged plumes. Clusters of data points at HCN concentrations below 0.75 ppbv are observed for the 18 August and 24 August data sets. For 18 August, there was no distinct BB influenced period and a minimal range in [HCN]. This led to the cluster of 18 August data points with $[O_3]$ near 60 ppbv. The 24 August data cluster near 50 pbbv $[O_3]$ captures the stable period after $[O_3]$ is depleted by ~7 ppbv upon smoke departure followed by a slow build in concentration.A similar figure with $O_3$ plotted against CO but for times specific to the arrival or departure of smoke is shown in the SI (Fig. S8).

Daily maximum P(ROx) values calculated from measured compounds ranged from 0.45 to 0.65 pptv s$^{-1}$ (Fig. 3). P(ROx) was dominated by HCHO photolysis and the reaction of O($^1$D) (from $O_3$ photolysis) with water vapor. ROx production from photolysis of acetaldehyde and acetone were of minor importance. Aldehydes were enhanced for some of the BB events. Typically, HCHO ranged between 1.5 and 4.5 ppbv, and BB events led to enhancements of near 2 ppbv for the 16, 17, and 24 August events. This led to distinct impacts on P(ROx) and is most evident for the 24 August event. Distinct enhancements in $O_3$ for the 17 August and 24 August events similarly affected P(ROx). While measurements of $[XO_2]$ generally reflect P(ROx) trends, this was not the case for the 17 August event. As $XO_2$ increased by ~27% upon the arrival of the smoke-affected air mass, P(ROx) from measured compounds increased by at most 5%. Changes in NOx were mostly negligible during this period, though [NO] remained below 0.1 ppbv indicating that small changes in [NO] could have had a large impact on $[XO_2]$. Increases in both $[O_3]$ and [HCHO] of ~20% did not contribute to a significant increase in P(ROx) due to a ~30% decrease in $[H_2O]$ (see Fig. S16 in the supplement) and a ~10% decrease in photolysis frequencies. The sudden increase in measured $[XO_2]$ when there were only small changes in P(ROx), NOx, and VOCs suggests the prominence of unmeasured ROx sources – most likely HONO. Peng et al. (2020) suggested HONO enhancement ratios ($\Delta HONO/\Delta CO$) of ~0.1 pptv ppbv$^{-1}$ for Western US wildfire BB plumes aged 3 hours, the likely age of our smoke plume sampled here, causing HONO photolysis to remain a significant ROx source even after 3 hours of aging. On average, HONO accounted for >90% and 50% of P(ROx) in 30 minute aged plumes and 3 hour aged plumes, respectively (Peng et al., 2020).

## 3.2 Ozone Production

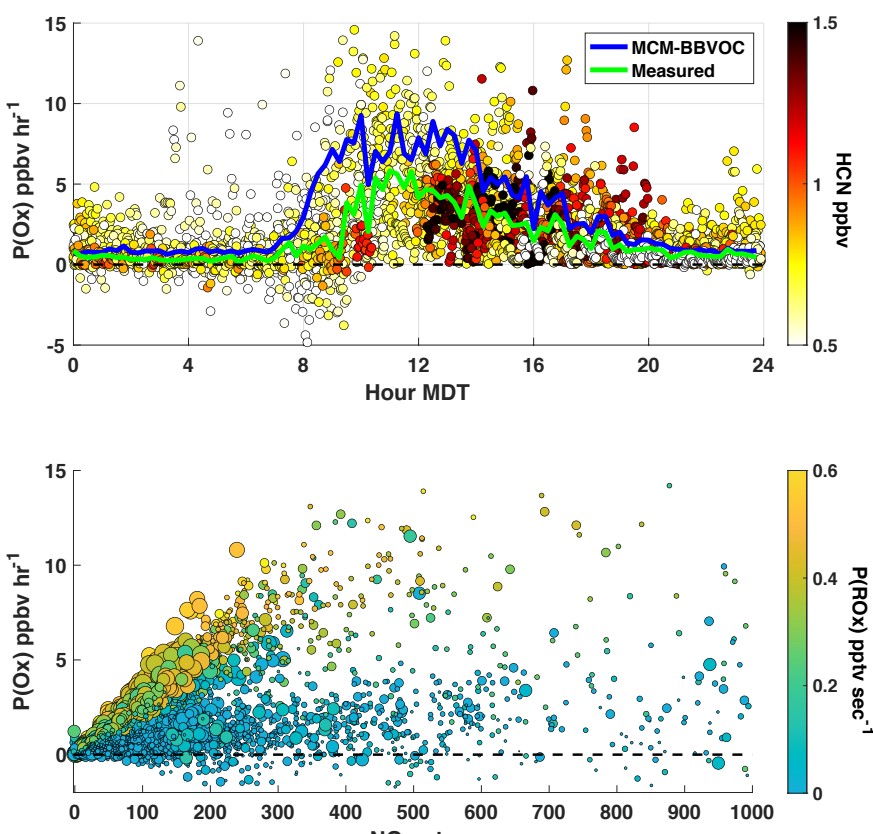

**Figure 5** P(Ox) results derived from 2 min XO$_2$ observations. Panel A shows the diurnal profile of all calculated P(Ox) values. The data points are colored by smoke tracer HCN. The green and blue traces represent 15-minute median values of P(Ox) as determined using measurements and model results, respectively. The provided model results were acquired with F0AM using the MCM-BBVOC mechanism. Panel B shows the variation of P(Ox) with NO. Data is colored by P(ROx) and sized by [HCN].

We describe the extent of ozone formation for the 16 August, 17 August and 24 August BB influenced periods using the commonly used $\Delta O_3/\Delta CO$ metric. These values depict the O$_3$ produced in transit to the McCall site while accounting for plume dilution or overall smoke influence of the site air sampled. $\Delta O_3/\Delta CO$ values were -0.02, 0.06 and 0.03 ppb ppbv$^{-1}$ for the 16, 17 and 24 August smoke events, respectively. These calculated values fall within the wide variability and range of literature $\Delta O_3/\Delta CO$ values for boreal and temperate forest fire smoke plumes aged less than two days, including numerous examples of ozone depletion for aged plumes (Jaffe and Wigder, 2012). Though the smoke was likely sourced from the same wildfire for the 16 August and 17 August events (section 2.4), we observe O$_3$ depletion on 16 August and O$_3$ enhancement on 17 August. $\Delta O_3/\Delta CO$ values were not calculated for the 22 August and 23 August smoke events as we were unable to attribute the observed increases of O$_3$ to smoke influence as they occurred at the same time that O$_3$ usually increased during non-smoky time periods as mentioned in the section 2.5. Ox enhancement ratios are not presented but differed insignificantly from O$_3$ enhancement ratios as NO$_2$ concentrations were much lower than O$_3$ concentrations (see Fig. 3).

Instantaneous O$_3$ production rates are calculated using NO and XO$_2$ concentrations (Fig. 3). Gaps in P(Ox) are due to measurement gaps in XO$_2$ when ECHAMP was offline for calibrations and diagnostic tests. The highest P(Ox) values occurred on 17 and 18 August during non-smoky periods between 10:00 MDT and 12:00 MDT, reaching formation rates slightly greater than 8 ppbv hr$^{-1}$. For the entire campaign, median P(Ox) peaked at 11:00 MDT at 5.8 ppbv hr$^{-1}$. As NO concentrations were low and rarely exceeded 1 ppbv, changes in [NO] had a near-linear impact on P(Ox). Noisy P(Ox) periods, such as the entire afternoon of 16 August, are mainly attributed to the atmospheric variability of and measurement precision for NO. Overall, there is little

correlation between P(Ox) and smoke tracers. However, elevated P(Ox) during the 17 August event is somewhat evident. The ~27% increase in $XO_2$ and near constant value for NO led to this temporary increase in P(Ox). P(Ox) increased from ~2.5 to 8.9 ppb hr$^{-1}$ during the transition from background air to significant smoke, remained elevated for 34 minutes, and then returned to near background P(Ox) rates. The overall lack of impact of BB influence on P(Ox) is further depicted in the P(Ox) diurnal cycle of Fig. 5. Modeled P(Ox) results for the same time period are also presented in Fig. 5a with the green median trend. These model results were acquired using F0AM with the MCM-BBVOC mechanism. Modeled P(Ox) is consistently greater than measured values with the greatest discrepancy occurring in the 7:45 to 8:15 MDT period. This difference is due to modeled [$XO_2$] being greater than measured [$XO_2$]. While we present results acquired using four unique chemical mechanisms, model predicted P(Ox) was always greater than measurements, though within the combined uncertainties.

Figure 5b shows the relationship between P(Ox), [NO], and P(ROx). For P(ROx) values above 0.4 ppt s$^{-1}$, P(Ox) increases almost linearly with [NO] up until at least 400 ppt, consistent with ozone formation being NOx-limited. The vast majority of NO concentrations are below 400 pptv. P(Ox) at lower P(ROx) values (less than 0.2 ppt/s) exhibits much more noise and are typically below 2 ppb/hr.

## 3.3 Model Evaluation

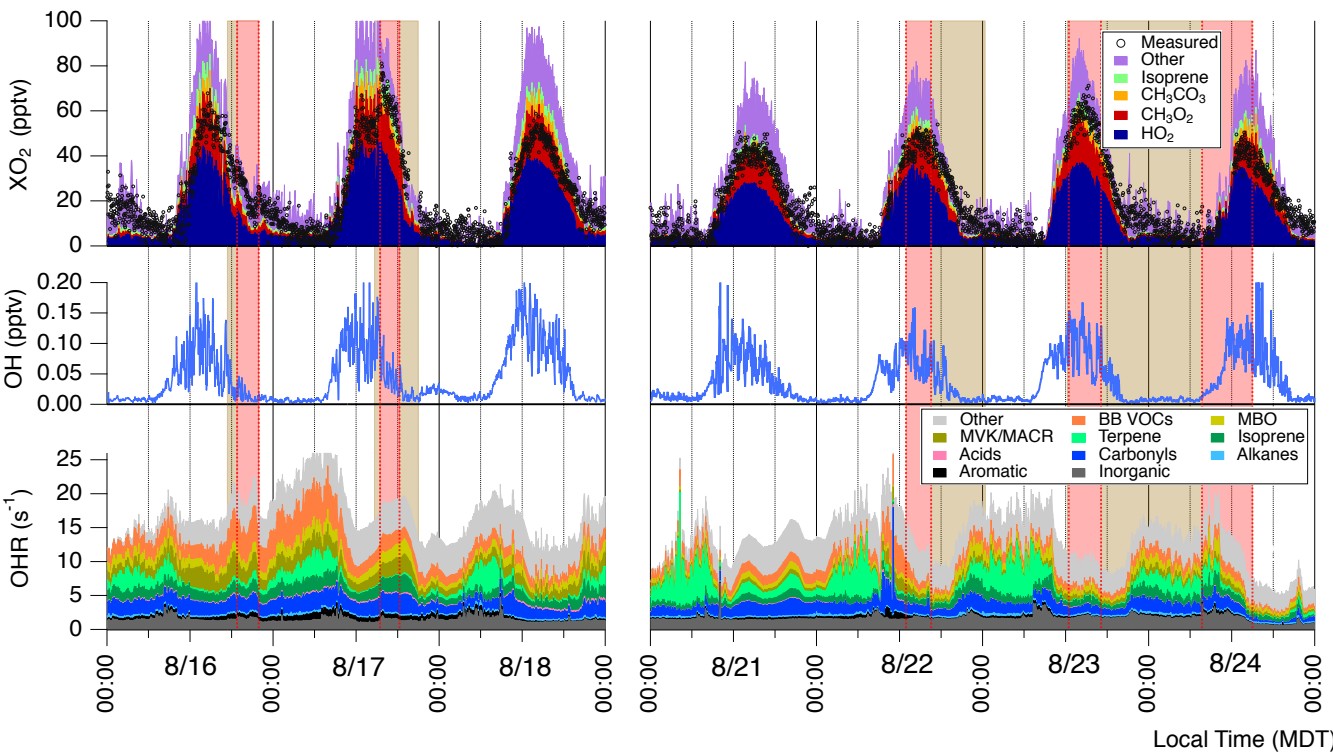

**Figure 6** Time series of modeled OH, $HO_2$, speciated $RO_2$, and OH reactivity (OHR). These results were acquired using the MCM-BBVOC mechanism. Periods of smoke are shaded as per Fig. 2. Measured $XO_2$ is included as black markers for comparison. Legend categories for $XO_2$ and OHR are mostly straightforward. Note that the isoprene $XO_2$ category includes several $RO_2$ species produced from isoprene oxidation.

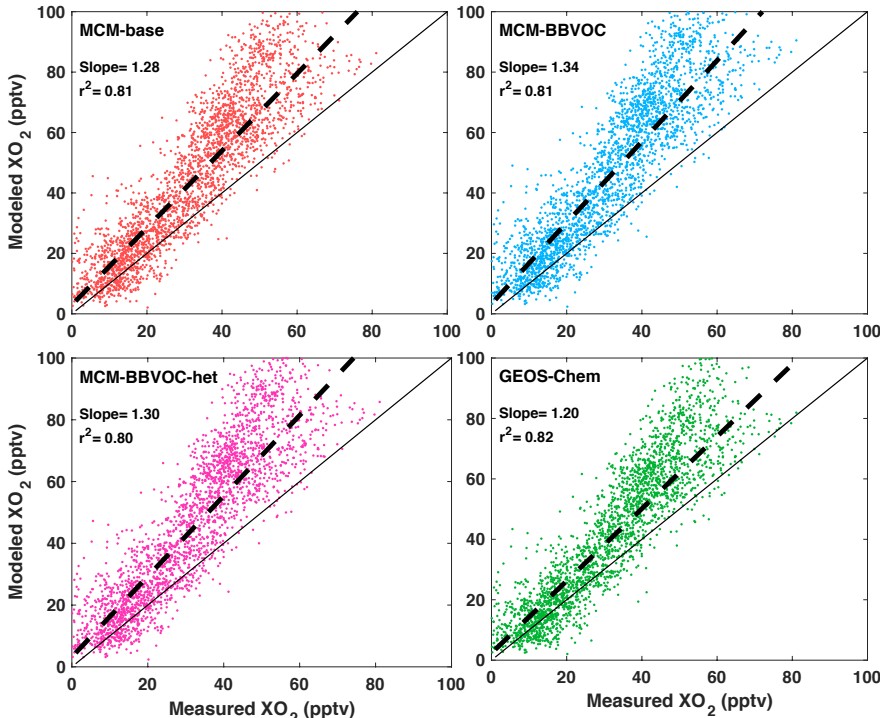

**Figure 7** Comparison of all daytime XO$_2$ observations (9:00 - 21:00 MDT) to modeled XO$_2$.

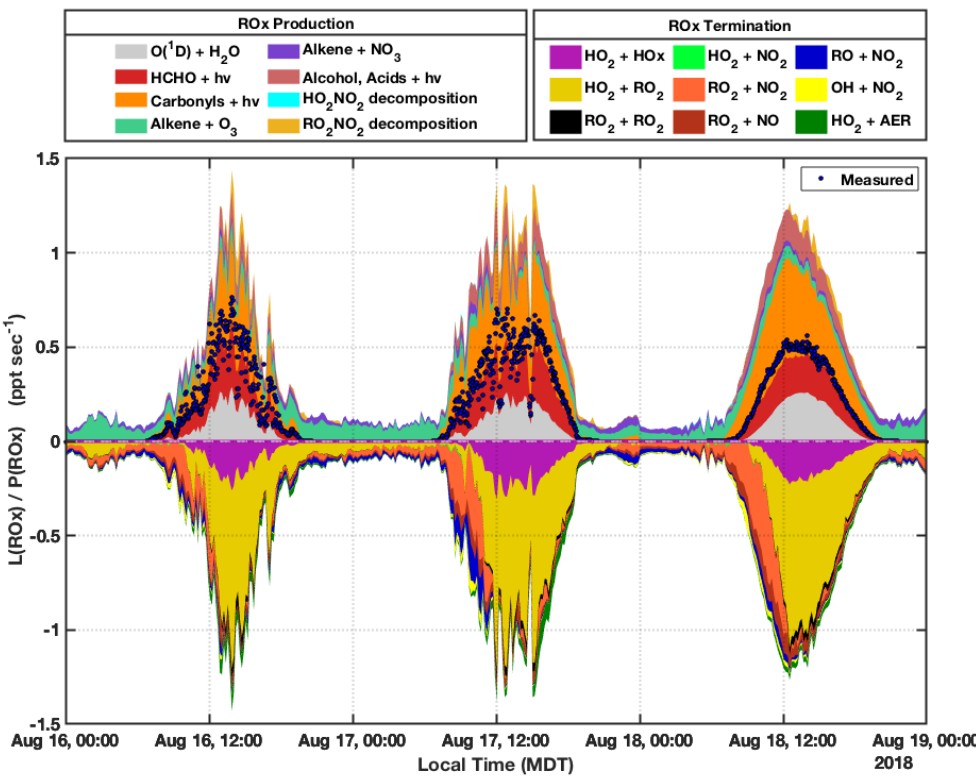

**Figure 8** Modeled ROx production and termination for 16, 17, and 18 August. The results provided were modeled with F0AM using the MCM-BBVOC-het mechanism. Net rates are provided for reversible processes. ROx termination L(ROx) and ROx production P(ROx) are separated into several straightforward categories as represented in their respective legends. Heterogeneous HO$_2$ loss is represented by the HO$_2$ + AER category. P(ROx) derived from direct measurements, as presented in Fig. 3, is shown for comparison and is represented by the blue markers. The modeled results are shown in 10-minute intervals and were averaged from the 2-minute basis, while measurement-based P(ROx) is shown in 2-minute intervals.

To investigate our understanding of photochemistry in biomass burning plumes we conducted zero-dimensional modeling constrained by our available measurements excluding ECHAMP $XO_2$ observations. Fig. 6 shows a time-series of model predicted $HO_2$, speciated $RO_2$, OH, and OH reactivity acquired using the MCM 3.3.1 mechanism with added BB VOC chemistry ("MCM-BBVOC"). Additional results included in the SI were acquired using the three other mechanisms: MCM-base, MCM-BBVOC-het, and the base GEOS-Chem mechanism (Figures S11-S13 in the supplement). Model [OH] (Fig. 6) had daily peak values ranging from 0.10 to 0.15 pptv ($2 - 3$ x $10^6$ molecules cm$^{-3}$). The MCM-base [OH] results were typically higher but generally agreed within 5% of the other two altered MCM mechanisms. Daytime GEOS-Chem OH concentrations were typically ~75% higher than the base MCM prediction. OH predictions were generally unaffected by smoke influence for all mechanisms, though upon smoke arrival on 17 August [OH] decreased from ~0.12 pptv to 0.05 pptv for the MCM-BBVOC mechanism with similar results observed with the other mechanisms. Modeled $XO_2$ comprises $HO_2$ (typically ~45-50%), $CH_3O_2$ (~20-25%), and the remaining portion a combination of $CH_3CO_3$, $RO_2$ derived from isoprene oxidation, and other organic peroxy radicals. Measured $XO_2$, as included in Fig. 3, is overlaid on Fig. 6. Daytime modeled [$XO_2$] (between 9:00 to 21:00 MDT) using the MCM-base, MCM-BBVOC, MCM-BBVOC-het, and GEOS-chem mechanisms are consistently greater than measured [$XO_2$] on average by 28%, 34%, 27%, and 20%, respectively (Fig. 7), all of which are within the combined $2\sigma$ measurement uncertainty (34%) and predicted model uncertainty (25%), though there are periods in which model values are greater than measured $XO_2$ by nearly a factor of two (i.e., the afternoon of 18 August). The inclusion of BB chemistry (MCM-BBVOC) increased $XO_2$ predictions (compared to MCM-base results), and the inclusion of heterogeneous $HO_2$ uptake (MCM-BBVOC-het) led to slight decreases (compared to MCM-BBVOC results). The rapid increase in [$XO_2$] observed on 17 August was not captured under any model conditions. As discussed in Sect. 2.6, this discrepancy from 17 August is likely at least partially due to the impact of HONO, which was not measured and unlikely properly accounted for by the zero-dimensional model. Observed $XO_2$ on 23 August was high compared to all other days observed, while $XO_2$ modeled is only somewhat greater than other days. Comparison between daytime $XO_2$ observations and model results (Fig. 7) was determined using the York bi-variate linear regression method (York et al., 2004). Both measured and modeled [$XO_2$] are low at night (below 10 pptv). Model $XO_2$ predictions with all chemical mechanisms agreed within 10%.

The OH reactivities from the MCM-BBVOC mechanism ranged from 5 to 25 s$^{-1}$ (Fig. 6). These OH reactivities are divided into several categories based on direct measurements and an 'other' category for non-measured model outputs. The 'other' category was often the greatest category contributing as much as 40 percent of the total reactivity at times. Incorporating heterogeneous losses to the BB VOC mechanism (MCM-BBVOC-het results) had nearly no effect on OH reactivity, and removing BB chemistry (MCM-base results) led to a smaller range in OH reactivity – from 5 to 20 s$^{-1}$. The GEOS-CHEM OH reactivities were lowest ranging 3 to 14 s$^{-1}$ due to this mechanism having a limited number of reactions causing fewer measured compounds to be constrained. The BB VOC category shown (catechol, furan, methyl furan, furfural, methyl furfural, and guaiacol) was often the greatest contributor to the portion of OH reactivity attributed to measured compounds. BB VOC values were typically 2 to 5 s$^{-1}$ accounting for 10 to 30% of measured reactivity. This category played a lesser role in the MCM-base mechanism as only one measured compound was included - catechol – leading to at most 10 percent of measured reactivity. Reactivity from carbonyls ($HCHO$, $CH_3CHO$, acetone, and methyl ethyl ketone (MEK)) typically contributed 15 to 20% of the measured reactivity. The inorganic category typically accounted for 10% of measured reactivity, or 1 to 3 s$^{-1}$, and was dominated by CO. The four biogenic categories included were isoprene, monoterpenes (measured monoterpenes sorted into alpha pinene, beta pinene, and limonene), MVK/MACR (methyl vinyl ketone and methacrolein), and MBO (2-methyl-3-buten-2-ol). These categories typically account for between 30 to 50% of the OH reactivity attributed to measured species. Other measured reactivity categories were minor and included aromatics (benzene, toluene, C2 benzenes, C3 benzenes, and phenol), alkanes ($CH_4$, $C_2H_6$, and ethyne), and acids (formic acid and acetic acid). The 'other' category had greatest contributions from carbonyl compounds. Changes in OH reactivity were

observed for some smoke periods. Subtle changes were noted during the 16 and 17 August events due to changes in the inorganic, carbonyl, biogenic, and BB VOC portions. A noticeable decrease in reactivity from 10 to <5 s$^{-1}$ occurred on 24 August upon smoke departure. Decreases in nearly all reactivity categories contributed with notable contributions due to depletion in aldehyde concentrations and [CO]. This period sustained the lowest reactivities for the entire data set while also having the lowest concentrations of smoke tracers.

Figure 8 shows modeled ROx production and termination acquired using MCM-BBVOC-het mechanism for the period of 16 August through 18 August. ROx production is sorted into several categories, and only net rates are provided for reversible processes (e.g., net decomposition of peroxy nitrates). The sum of measurement-based P(ROx), comprising the reaction of O($^1$D) with H$_2$O and the photolysis of HCHO, CH$_3$CHO, and acetone, is overlaid in Fig. 8 (shown speciated in Fig. 3) and typically accounts for between 50-60% of total modeled values. For modeled P(ROx), O($^1$D) + H$_2$O and HCHO photolysis both contribute ~25% of predicted ROx production during midday. Photolysis of carbonyls accounts for most of the remaining modeled daytime P(ROx), though only a fraction (less than 15%) of this category is from the measured carbonyl compounds acetaldehyde and acetone. Unmeasured carbonyls account for the rest, with methylglyoxal and glycolaldehyde accounting for 26% and 8%, respectively. Predicted methylglyoxal concentrations were typically between 0.4 and 0.6 ppbv. These concentrations are about an order of magnitude greater than those measured at mountaintop sites (Mitsuishi et al., 2018;Kawamura et al., 2013) but lower than those observed at a suburban site in China (Liu et al., 2020) and in biomass burning plumes observed in the Amazon (Kluge et al., 2020). Methylglyoxal is largely formed from the oxidation of MVK and MACR, themselves oxidation products of isoprene. A BB VOC related dicarbonyl, 4-oxo-2-pentenal (listed as C5DICARB within MCM), which is formed from methyl furan oxidation, accounted for 13% of P(ROx) from carbonyl photoloysis. Photolysis of acids and alcohols contribute up to ~10% of modeled P(ROx). Nighttime P(ROx) is typically ~0.1 pptv s$^{-1}$ and is primarily from alkene ozonolysis (> 80%) and the reaction of alkenes with NO$_3$. Net formation of peroxy RO$_2$NO$_2$, mainly PAN, was the dominant modeled ROx sink from the morning until ~ 12:00 MDT most days. Given that neither PAN nor the acetyl peroxy radical (CH$_3$CO$_3$) were directly measured and that a zero-dimensional model cannot be expected to accurately model PAN concentrations due to its long lifetime, this result is highly uncertain. Midday ROx termination was dominated by ROx self-reactions with ~75% of L(ROx) from HO$_2$ + RO$_2$ and the remainder of L(ROx) almost entirely from HO$_2$ + HO$_2$. Reactions between RO$_2$ and other RO$_2$ played a comparatively minor role (<5% L(ROx)). Other L(ROx) categories consisted of reactions of RO$_2$ + NO, RO + NO$_2$, and OH + NO$_2$. Heterogeneous uptake of HO$_2$ generally had small contributions to L(ROx) but at times of elevated PM concentrations such as the BB-influenced periods 17, 23, and 24 August, accounted for up to ~10% of total ROx termination (see section 3.4 for further discussion). Results from the 21 August through 24 August period (see Fig. S17 in SI) are similar to the results presented above for 16 – 18 August, though the unmeasured portion of P(ROx) is smaller for the former. This resulted from the considerably smaller concentrations of MVK, MACR, isoprene and BB VOCs measured during this period. The smaller portions for the carbonyls and 'alcohols, acids' categories led to smaller P(ROx) totals that peaked near 0.7 pptv s$^{-1}$ rather than the calculated 1.2 pptv s$^{-1}$ for the period shown in Fig. 8.

## 3.4 Model Sensitivity

There are important fundamental limitations to how well a zero-dimensional model can describe the McCall measurements. Concentrations of several radical precursors such as nitrous acid, glycolaldehyde, methylglyoxal, and glyoxal were not constrained by measurements and were instead determined by the model. For days in which dilute biomass burning plumes arrived suddenly, it is unrealistic to expect that the model can accurately determine the concentrations of these compounds which depend on the history of the air mass. XO$_2$ predictions are sensitive to several model inputs. Since some secondary compounds'

concentrations build up in the model and can drive significant ROx production, we explore the sensitivity of model $XO_2$ predictions to the first-order dilution rate constant applied to all compounds. Shortening the dilution lifetime from 24-hr to 6-hr (the minimum value suggested by Wolfe et al. (2016) decreases model $XO_2$ predictions and reduces the GEOS-Chem $XO_2$ overprediction from 20% to 2% (Fig. S15 in SI). The most important unmeasured radical precursors that are affected by this dilution are methylglyoxal, glycolaldehyde, and 4-oxo-2-pentenal due to their collective contribution to P(ROx). We also investigate model $XO_2$ sensitivity to NOx. Increasing NOx inputs by 50% leads to peak daily $XO_2$ predictions by roughly 10%.

As the 15 pptv increase in $[XO_2]$ observed on 17 August is not captured by model simulations, we include HONO as an additional model constraint in additional GEOS-Chem simulations (see Sect. S8 of SI). Constrained HONO concentrations were determined by the product of selected $\Delta$HONO/$\Delta$CO values to the measured CO mixing ratios during BB periods. To achieve a similar ~15 pptv $XO_2$ enhancement as measured during the 17 August BB event, a HONO enhancement ratio of near 3 pptv ppbv$^{-1}$ is required, which provides an additional 0.15 - 0.60 ppbv HONO throughout the BB period. This $\Delta$HONO/$\Delta$CO value is 30 times larger than observed by Peng et al. (2020) for similarly aged plumes. While this value is likely unrealistic, larger $\Delta$HONO/$\Delta$CO have been reported by Peng et al. (2020). Other unmeasured ROx precursors were likely present and at least partially responsible for the elevated $XO_2$ concentrations observed.

Model sensitivity to heterogeneous $HO_2$ uptake was also investigated. The introduction of heterogeneous losses of $HO_2$ had overall minimal impacts (see Fig. 7 MCM-BBVOC-het results) even though a fairly high $HO_2$ uptake coefficient of 0.2 was used (Abbatt et al., 2012). Heterogeneous losses decreased modeled [OH] and $[XO_2]$ by an average of 3.4% and 2.9%, respectively, though the impact is more evident for smoke periods with elevated PM. Our analysis of $HO_2$ heterogeneous uptake is limited by the uncertainty in the $HO_2$ uptake coefficient and the specific surface area parameter. To investigate the sensitivity of model results to these parameters, these parameters were varied for several GEOS-Chem model simulations (see Figures S9 and S10 in SI). We focus our heterogeneous chemistry sensitivity tests on a 40 min period during the 17 August BB event in which OA concentrations were 30 µg m$^{-3}$. Inclusion of heterogeneous chemistry with standard parameter settings ($\gamma = 0.2$) leads to a 11% decrease in $[HO_2]$, whereas use of a much higher and likely unrealistic $HO_2$ uptake coefficient of 0.5 resulted in a 25% $HO_2$ decrease. A similar $HO_2$ decrease results from using a higher specific surface area of 10 m$^2$ g$^{-1}$. Use of a smaller $HO_2$ uptake coefficient of 0.02 led to a nearly negligible decrease in $[HO_2]$. Constraining the OA concentration at 100 µg m$^{-3}$ for the same period - much higher than actually observed – leads to heterogeneous $[HO_2]$ loss of 3% for an uptake coefficient of 0.02 and 30% for an uptake coefficient of 0.2. The only conditions in which heterogeneous loss of $HO_2$ to BB smoke would appear to be important would be for less dilute plumes ($[OA]] > 100$ µg m$^{-3}$) and a high uptake coefficient ($\gamma > 0.2$).

## 4 Conclusions

Peroxy radicals were measured during the FIREX 2018 campaign in McCall, Idaho in order to better characterize ROx chemistry and study $O_3$ formation within BB plumes. There were five distinct BB influenced periods that were identified using smoke tracers, primarily HCN. HYSPLIT back trajectories were paired with satellite data to suggest smoke sources and plume age. Most smoke periods had distinct enhancements in $O_3$ with enhancement ratios of up to 0.06 ppbv ppbv$^{-1}$ $\Delta O_3$/$\Delta$CO. Zero-dimensional box model results for $XO_2$ were generally greater than measured $XO_2$. These simulated results were acquired with F0AM using a variety of chemical mechanisms - GEOS-Chem, MCM, and two expanded versions of MCM. All model iterations followed the general trends observed for $[XO_2]$ measurements, though a measured 15 pptv $XO_2$ enhancement during a 17 August BB event was not captured by any model iteration. This includes simulated results acquired using expanded versions of the MCM that had additional BB VOC chemistry and heterogeneous $HO_2$ and OH losses intended to better capture BB influence.

Heterogeneous losses overall had minimal impact on $XO_2$ even though BB smoke periods with elevated organic PM levels often near 30 μg m³ led to deceases in $[HO_2]$ by 10%. Heterogeneous chemistry was investigated using a variety of $HO_2$ uptake coefficients and aerosol specific surface areas. At the greatest settings, these variables led to decreases in $HO_2$ of at most 25% and 25%, respectively, during a period with PM above 30 μg m⁻³. Heterogeneous $HO_2$ uptake is likely a major $HO_2$ sink and thus has a large impact on HOx chemistry for less dilute BB plumes. Both measured and modeled $XO_2$ concentrations were used to calculated P(Ox). The presence of smoke had an overall negligible impact on P(Ox) as NOx enhancements were minimal.

Quantification of P(ROx) suffered from lack of constraints for several compounds including HONO, methylglyoxal, and glycolaldehyde. HONO photolysis likely contributed to the enhanced concentrations in $[XO_2]$ measured during the 17 August BB period, though a rather large ΔHONO/ΔCO value near 3.0 ppbv ppbv⁻¹ is necessary for a similar $XO_2$ enhancement. An additional gas-phase process in need of validation by measurements is the role of PAN formation as a ROx sink during the morning. The importance of unmeasured ROx precursors was especially sensitive to the first-order dilution rate constant applied to all unmeasured species. Finally, the role of heterogeneous $HO_2$ uptake as a ROx sink would benefit from more direct measurements of particle size distribution and knowledge of $HO_2$ uptake coefficients. Heterogeneous $HO_2$ uptake was minimal for the dilute BB plumes studied here, and it would appear to only be important in less dilute BB plumes if the uptake coefficient is relatively high (e.g., 0.2).

## Data Availability

Peroxy radical measurements are available at https://doi.org/10.26023/CY1Q-QT7V-G80R. Other supporting measurements are available at https://data.eol.ucar.edu/master_lists/generated/we-can/ and upon request.

## Supplement

## Author contributions

AJL and ECW wrote the manuscript. All authors discussed results and commented on the manuscript. AJL, ECW, and DCA were responsible for $XO_2$ measurements. JRR, CD, and SCH operated and were responsible for the $O_3$ instrument and the TILDAS instruments for trace gases. FM, JEK, and WBK contributed with VOCUS measurements. RAW, YL, and AHG contributed with cTAG VOC measurements. EF and PLC were responsible for particle phase measurements. TIY and SCH coordinated the ARI data analysis.

## Competing Interests

The authors declare that they have no conflict of interest.

## Acknowledgements

This work was supported by NOAA/ACCC grants NA16OAR4310105 to Drexel University, NA16OAR4310104 to Aerodyne Research, Inc., and NA16OAR4310107 to UC Berkeley.

We profoundly thank Michele Crester and the Brundage Mountain Company for the use of the Activity Barn in McCall, Idaho as our sampling location. We acknowledge the work of additional scientists Leonid Nichman and Nathan Kreisberg for their operation and support of instruments in the field. We thank Eben Cross of QuantAQ, Inc. in Somerville, MA for providing ARISense solar irradiance data. We also acknowledge Jason Curry, Connor Daube, and Salvador Cartagena of Aerodyne Research Inc. for their
general assistance including the transport of the mobile laboratories.

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
