# Peer review of "Ground-based Investigation of HOx and Ozone Chemistry in Biomass Burning Plumes in Rural Idaho"

_Atmospheric Chemistry and Physics, 2021_

## Author Comment (AC2)

**Point-by-point Response to Reviewer's Comments**

We appreciate the three reviewers for taking time to carefully review the manuscript and give detailed and constructive comments, which has greatly helped to improve this paper. Below is our point-by-point response to each comment.

Responses to the reports of Anonymous Referees #2 and #3 begin on page 11 and page 19, respectively. Response to community comments is included on page 24.

**Report #1 by Anonymous Referee #1**

1.) In the manuscript, the discrepancies between measured and modeled [XO2] are noted, but not fully explained. Namely, what causes the over-estimate of XO2 in models? Any recommendations to the chemical mechanisms? Or the difference is within the measurement uncertainty, which undermines the valuableness of the XO2 measurement? Looking at Figure 6, model sometimes is 50-60% higher than measurement. Why? The difference shown in Figure 6 seems larger than the 31% noted in the abstract. In fact, how is the "31%" calculated?

**Response:** We agree that the original manuscript had insufficient discussion of the model results and we have greatly expanded this portion of the manuscript in the revision. The model overestimates of peroxy radical concentrations appear to be mainly from the build-up of unmeasured photolabile carbonyls/acids within the model. To investigate this we have conducted additional model simulations and the results are discussed within a new section "Model Sensitivity" (Sect. 3.4) with additional figures added to the SI (Figures S14 – S15). We do not recommend any additional reactions to chemical mechanisms, though we suggest the potential importance of heterogeneous chemistry for concentrated smoke plumes. We find the model predictions to be sensitive to the parameters that control the first-order dilution of unmeasured species (Fig. S14), and recommend that measurements of glyoxal, methylglyoxal, and glycolaldehyde become more commonplace.

While the difference between the model and measured values falls within the combined uncertainties as stated within the text (Lines 444-447), we respectfully disagree that this undermines the value of XO2 measurements, which are still rare and difficult. While model predictions and estimates are quite useful, they themselves include large errors. As shown in the new *Model Sensitivity* section, adjusting key parameters within the zero-dimensional model used can lead to outputs that vary by more than 30%. Similarly, the Leighton method, which is based on the assumption of photostationary state, is extremely sensitive to the accuracy of the measurements of NO, NO2, O3, and jNO2, with uncertainties that sometimes exceed 100%. Models and estimates benefit greatly by verification by direct measurements, which we attempt in the presented research. We agree that the value of model-measurement comparisons for ROx compounds will greatly benefit from reduced uncertainties in the measurement methods, which currently are largely limited by the accuracy of calibration methods.

Finally, the last comment regarding 31% stated within the abstract was initially misstated. This percentage, which should have been listed as 34% instead, is based on the average daytime regression determined using Fig. 7 (model XO2 vs measured XO2). We revised the abstract text to improve clarity by stating that the model predictions were "high by an average of". Rather than stating the highest overpredictions (models that include additional BB oxidation chemistry), we state the average over-predictions for the base mechanisms of MCM and GEOS-Chem.

**Revised Text:**

New section: Sect. 3.4 Model Sensitivity

The final paragraph is mostly integrated from text previously included in the model P(ROx) discussions, though some text has remained in the overall modeled P(ROx) discussion with some new text added here.

[revised manuscript text omitted]

Additional SI Figures: Here we include Fig. 15 along with its caption. We include Figure S14 in a subsequent comment that is more directly related to that figure (model simulations using HONO constraints from selected  $\Delta$ HONO/ $\Delta$ CO)

**Figure S15.** F0AM XO2 predictions with varied first-order dilution rate constants ( $k_{Dil}$ ) obtained using the GEOS-Chem mechanism. ECHAMP XO2 measurements are included for comparison. Model predicted XO2 results are shown for simulations that used the standard 24-hr dilution lifetime ( $k_{Dil} = 1/86400 \text{ s}^{-1}$ ), a 6-hr dilution lifetime ( $k_{Dil} = 4/86400 \text{ s}^{-1}$ ), and no dilution ( $k_{Dil} = 0 \text{ s}^{-1}$ ).

Relevant text within SI for Fig. S15:

The sensitivity of model predicted XO2 to the first-order dilution rate constant (kDil) was investigated using additional F0AM results acquired with the base GEOS-Chem mechanism (Fig. S15). Completely removing dilution allows for the further buildup of secondary species and higher predictions of [XO2]. Increasing the rate constant from the standard 24-hr lifetime (kDil = 1/86400 s-1) to a 6-hr lifetime (kDil = 4/86400 s-1) leads to decreased XO2 predictions that better align with the measured XO2 values. The most important unmeasured radical precursors that are affected by this dilution setting are methyl glyoxal and glycolaldehyde. For results acquired using the MCM-BBVOC mechanism, the dilution

setting is important for additional processes including limiting the buildup of photolabile carbonyls such as 4-oxo-2-pentenal (listed as C5DICARB within MCM).

2.) The effect of HO2 heterogeneous uptake on the radical budget is likely over-stated. As the authors noted, the HO2 uptake coefficient is highly uncertain. A relatively large value (i.e., 0.2) is chosen. In dense BB plumes, organic aerosol is the dominant composition with mass fraction up to 80%. As discussed in Abbatt et al. 2012 and George et al., the HO2 uptake coefficient to solid organic particles is < 0.001 and to liquid organic particles is < 0.01. If applying the small gamma values, the HO2 heterogeneous is negligible even in dense BB plumes.

**Response:** We stated in our initial text that we used a fairly large uptake coefficient, so we were immediately in some agreement with this comment. Since we have already varied our uptake coefficients used for several model simulations (now shown in Fig. 9 of SI), we added additional text to clarify this issue. We now state that model results acquired using an uptake coefficient of 0.02 leads to near negligible differences when compared to models that include no heterogeneous chemistry. For our statement where we extrapolate the impact of heterogeneous chemistry to more concentrated plumes (> 100 ug m-3), we restate that we have used a fairly high uptake coefficient to underscore the uncertainty.

**Revised Text** The original paragraph focusing on heterogeneous chemistry has been mostly reworked into the new 'Model Sensitivity' Sect. 3.4 as its final paragraph (see full revised text above on page 2). This paragraph (lines 535-547) is nearly entirely revised. Included here is the most relevant text of this revised paragraph:

We focus our heterogeneous chemistry sensitivity tests on a 40 min period during the 17 August BB event in which OA concentrations were 30  $\mu$ g m-3. Inclusion of heterogeneous chemistry with standard parameter settings ( $\gamma = 0.2$ ) leads to a 11% decrease in [HO2], whereas use of a much higher and likely unrealistic HO2 uptake coefficient of 0.5 resulted in a 25% HO2 decrease. A similar HO2 decrease results from using a higher specific surface area of 10 m2 g-1. Use of a smaller HO2 uptake coefficient of 0.02 led to a nearly negligible decrease in [HO2]. Constraining the OA concentration at 100  $\mu$ g m-3 for the same period - much higher than actually observed – leads to heterogeneous [HO2] loss of 3% for an uptake coefficient of 0.02 and 30% for an uptake coefficient of 0.2. The only conditions in which heterogeneous loss of HO2 to BB smoke would appear to be important would be for less dilute plumes ([OA]] > 100  $\mu$ g m-3) and a high uptake coefficient ( $\gamma > 0.2$ ). Text in the conclusion has been revised (line 573). The revised portion is highlighted.

Finally, the role of heterogeneous HO2 uptake as a ROx sink would benefit from more direct measurements of particle size distribution and knowledge of HO2 uptake coefficients. Heterogeneous HO2 uptake was minimal for the dilute BB plumes studied here and would appear to only be important in less dilute BB plumes if the uptake coefficient is relatively high (e.g., 0.2).

3.) Missing HONO as model input is invoked in several places as a possible reason for the model vs measurement difference, but this reasoning is questionable. First of all, the delta\_HONO/delta\_CO after 3hr aging of BB plume, as shown in Peng et al., is 0.1 ppt/ppb, rather than 1 ppt/ppb as quoted in the manuscript (Line 377). In addition, Peng et al. clearly stated that after 3hr the [HONO] is near or below the instrument detection limit and they refrain from interpreting what it implies in terms of potential HONO steady state in aged plumes. Lastly, HONO can be

added to the box model (using delta\_HONO/delta\_CO = 0.1 ppt/ppb and measured CO), to directly test the effect of missing HONO on the modeling results.

**Response:** We agree that further investigation into HONO as a potential HOx source would improve this manuscript. We have updated the typo (1.0 pptv ppbv-1 vs 0.1 pptv ppb-1) that all reviewers identified. We have included additional model simulations where we constrain HONO using CO data and selected values of  $\Delta$ HONO/ $\Delta$ CO, following the suggestion of Anonymous Referees #1 and #2. Due to the large discrepancy between measured and modeled XO2 during the 17 August BB event, we specifically investigate HONO as a missing HOx precursor for this event. Simulations were run using a variety of HONO constraints (a range in  $\Delta$ HONO/ $\Delta$ CO were used), and resulting XO2 results for 17 August have been included in the SI in Fig. S14. We find a rather large  $\Delta$ HONO/ $\Delta$ CO of 3.0 pptv ppbv-1 is required for a similar XO2 enhancement as measured. Additional text was added within our 'Model Sensitivity' Sect. 3.4 and is included below. We also include text added to the SI regarding the new Fig. S14. Relevant text in the SI and conclusion have also been updated accordingly.

**Revised Text (Sect. 3.4, Lines 529-536):**

As the 15 pptv increase in  $[XO_2]$  observed on 17 August is not captured by model simulations, we include HONO as an additional model constraint in additional GEOS-Chem simulations (see Sect. S8 of SI). Constrained HONO concentrations were determined by the product of selected  $\Delta$ HONO/ $\Delta$ CO values to the measured CO mixing ratios during BB periods. To achieve a similar ~15 pptv XO2 enhancement as measured during the 17 August BB event, a HONO enhancement ratio of near 3 pptv ppbv-1 is required, which provides an additional 0.15 - 0.60 ppbv HONO throughout the BB period. This  $\Delta$ HONO/ $\Delta$ CO value is 30 times larger than observed by Peng et al. (2020) for similarly aged plumes. While this value is likely unrealistic, larger  $\Delta$ HONO/ $\Delta$ CO have been reported by Peng et al. (2020). Other unmeasured ROx precursors were likely present and at least partially responsible for the elevated XO2 concentrations observed.

**Additional SI Figure:**

**Figure S14.** F0AM XO2 predictions with constrained HONO for 17 August period. Model results were obtained using the base version of GEOS-Chem. The initial HONO data constrained is based on the model outputs of HONO acquired using GEOS-Chem. Enhancements in HONO are then added for the 17 August BB event at 15:27 MDT using HONO enhancement ratios ( $\Delta$ HONO/ $\Delta$ CO).

Relevant text within SI for Fig. S15:

To investigate the importance of HONO in the model results, we ran additional F0AM simulations using the GEOS-Chem mechanism with HONO concentrations constrained using a range of  $\Delta$ HONO/ $\Delta$ CO values and the measured CO mixing ratios. The raw enhancements in [HONO] for this event were 0.005 – 0.020 ppbv, 0.050 – 0.200 ppbv, and 0.150 – 0.600 ppbv when the  $\Delta$ HONO/ $\Delta$ CO value was set as 0.1 pptv ppbv-1, 1.0 pptv ppbv-1, 3.0 pptv ppbv-1, respectively. XO2 predictions for the 17 August BB event are shown in Fig. S14. In order for the model to produce the 15 pptv measured increase in [XO2] we need to constrain HONO during BB influenced period using a rather large  $\Delta$ HONO/ $\Delta$ CO of 3.0 pptv ppbv-1 – much greater than the  $\Delta$ HONO/ $\Delta$ CO ratios observed for similarly aged plumes by Peng et al. (2020). While this value is likely unrealistic, larger  $\Delta$ HONO/ $\Delta$ CO have been reported by Peng et al. (2020) and can be implied up to 5.9 pptv ppbv-1, though for considerably younger smoke plumes, using temperate forest emission factors (Akagi et al., 2011).

Additional Changes to Abstract and Conclusion: Since the HONO enhancement ratio is rather large, we

remove its mention from the abstract and replace with only 'unmeasured HOx sources'. Our statement regarding

that 17 August event within the abstract has been updated:

One period of BB influence had distinct measured enhancements of 15 pptv XO2 that were not reflected in the model output, likely due to the presence of unmeasured HOx sources.

Within the conclusion we now state the  $\Delta HONO/\Delta CO$  value required for the 15 pptv rather than invoking its

potential as a HOx precursor during the 17 August event (lines 565-566):

HONO photolysis likely contributed to the enhanced concentrations in [XO2] measured during the 17 August BB period, though a rather large  $\Delta$ HONO/ $\Delta$ CO value near 3.0 ppbv ppbv-1 is necessary for a similar XO2 enhancement.

4.) Figure 5 is interesting and related discussions should be expanded. For example, what does the lack of correlation between P(O3) and NO when P(ROx) is small suggest. Does it suggest O3 formation is VOC-limited? More information can be distilled from the figure, or from the four variables in the figure (P(O3), P(ROx), NO, and HCN).

**Response:** We do not feel comfortable commenting on the lack of correlation in Fig. 5b ( $P(O_3)$  vs NO) due to the noise associated in our  $P(O_3)$  calculation. At low P(ROx) values, P(Ox) is low - typically less than 2 ppb hr-1. The large variability in P(Ox) for this subset of data precludes making any conclusions regarding its sensitivity to [NOX].

5.) In the manuscript, some metrics are from direct measurement (e.g., [XO2]), some metrics are calculated from measures species (e.g., P(Ox) from Eqn. (3)), and some metrics are purely based on box model. These need to

carefully worded in the discussion to avoid confusion. For example, Line 460 and others use the term "measured OH reactivity", which the reader believe is the calculated OH reactivity based on measured VOCs. Then, Line 477 refers to Eqn. (3) as measured P(ROx), which makes the reader confused for a bit, as it is calculated, not measured. Such subtleties hinder the readability of the manuscript.

**Response:** We agree with this issue regarding the clarity of wording for metrics calculated from direct measurements. This has been addressed throughout the paper. There are two instances in which "measured OH reactivity" was mentioned (now lines 473 and 780). This wording has been changed to "OH reactivity attributed to measured compounds". The term "measured P(ROx)" has been changed to 'measurement-based P(ROx)'.

6.) Line 482. Are there any constraints or independent verification on the contribution of photolysis of methyl glyoxal and glycolaldehyde to the P(ROX)? Their contributions seem much larger than expected, based on measurements of these two species in previous wildfire studies.

**Response:** The contributions of methyl glyoxal was initially understated, and the text has been updated to state that the methyl glyoxal contributes 26% (previously listed as 19%) to the dominant 'carbonyl + hv' P(ROx) category. Further discussion focusing on the significant carbonyls contributing to this P(ROx) category has been added. We focus on methyl glyoxal, glycolaldehyde, and a dicarbonyl formed from BB VOC chemistry. We have included text stating that the precursors to methyl glyoxal and glycolaldehyde are from the constrained compounds of MVK/methacrolein and 2-MBO, respectively. We add context to this issue by listing concentrations of important species, and for methyl glyoxal we compare our modeled concentrations to values measured by other studies.

**Revised Text:**

Relevant text within our model-based P(ROx) discussion (lines 493-501) is provided with newly added lines highlighted.

Photolysis of carbonyls accounts for most of the remaining modeled daytime P(ROx), though only a fraction (less than 15%) of this category is from the measured carbonyl compounds acetaldehyde and acetone. Unmeasured carbonyls account for the rest, with methylglyoxal and glycolaldehyde accounting for 26% and 8%, respectively. Predicted methylglyoxal concentrations were typically between 0.4 and 0.6 ppby. These concentrations are about an order of magnitude greater than those measured at mountaintop sites (Mitsuishi et al., 2018;Kawamura et al., 2013) but lower than those observed at a suburban site in China (Liu et al., 2020) and in biomass burning plumes observed in the Amazon (Kluge et al., 2020). Methylglyoxal is largely formed from the oxidation of MVK and MACR, themselves oxidation products of isoprene. A BB VOC related dicarbonyl, 4-oxo-2-pentenal (listed as C5DICARB within MCM), which is formed from methyl furan oxidation, accounted for 13% of P(ROx) from carbonyl photoloysis.

**Minor comments from Anonymous Referee #1:**

Line 26-28. The reader suggests to rephrase the sentence to "the model over-estimates the XO2 by 30%".

**Response:** We have revised the abstract. The phrasing of "the model agreed within 31%" has been changed to more closely reflect this suggestion.

**Revised abstract text (Lines 28-29):**

The models consistently over-estimated  $XO_2$  with the base MCM and GEOS-Chem  $XO_2$  predictions high by an average of 28% and 20%, respectively.

Line 29-30. Suggest rephrasing to "likely due to the presence of an unmeasured HOx source that is not included in models". The whole sentence may need to be rewritten after investigating the role of HONO as mentioned above. **Response:** Since we have found that rather a large HONO enhancement ratio was required for modeling a similar XO2, we have updated the sentence to more closely resemble the above suggestion.

**Revised Abstract Text (Lines 29-30):**

One period of BB influence had distinct measured enhancements of 15 pptv XO2 that were not reflected in the model output, likely due to the presence of unmeasured HOx sources.

Figure 5b. P(O3) is used in the y-axis, but P(Ox) is used in the figure caption. Be consistent.

**Response:**  $P(O_3)$  within the figure and any mentions in the text related to Figure 5b (lines 499-502) have been updated to P(Ox). A similar comment from Anonymous Reviewer #2 suggested introducing a legend to figure 5a. The revised figure is presented here.

**Revised Figure 5:** Figure 5a with an included legend. Figure 5b with ' $P(O_3)$ ' replacing ' $P(O_3)$ ' of y-axis title.

---

## Author Comment (AC3)

**Point-by-point Response to Reviewer's Comments**

We appreciate the three reviewers for taking time to carefully review the manuscript and give detailed and constructive comments, which has greatly helped to improve this paper. Below is our point-by-point response to each comment.

Responses to the reports of Anonymous Referees #2 and #3 begin on page 11 and page 19, respectively. Response to community comments is included on page 24.

**Report #1 by Anonymous Referee #1**

1.) In the manuscript, the discrepancies between measured and modeled [$XO_2$] are noted, but not fully explained. Namely, what causes the over-estimate of $XO_2$ in models? Any recommendations to the chemical mechanisms? Or the difference is within the measurement uncertainty, which undermines the valuableness of the $XO_2$ measurement? Looking at Figure 6, model sometimes is 50-60% higher than measurement. Why? The difference shown in Figure 6 seems larger than the 31% noted in the abstract. In fact, how is the "31%" calculated?

**Response:** We agree that the original manuscript had insufficient discussion of the model results and we have greatly expanded this portion of the manuscript in the revision. The model overestimates of peroxy radical concentrations appear to be mainly from the build-up of unmeasured photolabile carbonyls/acids within the model. To investigate this we have conducted additional model simulations and the results are discussed within a new section "Model Sensitivity" (Sect. 3.4) with additional figures added to the SI (Figures S14 – S15). We do not recommend any additional reactions to chemical mechanisms, though we suggest the potential importance of heterogeneous chemistry for concentrated smoke plumes. We find the model predictions to be sensitive to the parameters that control the first-order dilution of unmeasured species (Fig. S14), and recommend that measurements of glyoxal, methylglyoxal, and glycolaldehyde become more commonplace.

While the difference between the model and measured values falls within the combined uncertainties as stated within the text (Lines 444-447), we respectfully disagree that this undermines the value of $XO_2$ measurements, which are still rare and difficult. While model predictions and estimates are quite useful, they themselves include large errors. As shown in the new *Model Sensitivity* section, adjusting key parameters within the zero-dimensional model used can lead to outputs that vary by more than 30%. Similarly, the Leighton method, which is based on the assumption of photostationary state, is extremely sensitive to the accuracy of the measurements of NO, $NO_2$, $O_3$, and $j_{NO2}$, with uncertainties that sometimes exceed 100%. Models and estimates benefit greatly by verification by direct measurements, which we attempt in the presented research. We agree that the value of model-measurement comparisons for ROx compounds will greatly benefit from reduced uncertainties in the measurement methods, which currently are largely limited by the accuracy of calibration methods.

Finally, the last comment regarding 31% stated within the abstract was initially misstated. This percentage, which should have been listed as 34% instead, is based on the average daytime regression determined using Fig. 7 (model $XO_2$ vs measured $XO_2$). We revised the abstract text to improve clarity by stating that the model predictions were "high by an average of". Rather than stating the highest overpredictions (models that include additional BB oxidation chemistry), we state the average over-predictions for the base mechanisms of MCM and GEOS-Chem.

**Revised Text:**

New section: Sect. 3.4 Model Sensitivity

The final paragraph is mostly integrated from text previously included in the model P(ROx) discussions, though some text has remained in the overall modeled P(ROx) discussion with some new text added here.

[revised manuscript text omitted]

Additional SI Figures: Here we include Fig. 15 along with its caption. We include Figure S14 in a subsequent comment that is more directly related to that figure (model simulations using HONO constraints from selected $\Delta HONO/\Delta CO$)

[Figure]

**Figure S15.** F0AM $XO_2$ predictions with varied first-order dilution rate constants ($k_{Dil}$) obtained using the GEOS-Chem mechanism. ECHAMP $XO_2$ measurements are included for comparison. Model predicted $XO_2$ results are shown for simulations that used the standard 24-hr dilution lifetime ($k_{Dil} = 1/86400$ s$^{-1}$), a 6-hr dilution lifetime ($k_{Dil} = 4/86400$ s$^{-1}$), and no dilution ($k_{Dil} = 0$ s$^{-1}$).

Relevant text within SI for Fig. S15:

The sensitivity of model predicted $XO_2$ to the first-order dilution rate constant ($k_{Dil}$) was investigated using additional F0AM results acquired with the base GEOS-Chem mechanism (Fig. S15). Completely removing dilution allows for the further buildup of secondary species and higher predictions of $[XO_2]$. Increasing the rate constant from the standard 24-hr lifetime ($k_{Dil} = 1/86400$ s$^{-1}$) to a 6-hr lifetime ($k_{Dil} = 4/86400$ s$^{-1}$) leads to decreased $XO_2$ predictions that better align with the measured $XO_2$ values. The most important unmeasured radical precursors that are affected by this dilution setting are methyl glyoxal and glycolaldehyde. For results acquired using the MCM-BBVOC mechanism, the dilution

setting is important for additional processes including limiting the buildup of photolabile carbonyls such as 4-oxo-2-pentenal (listed as C5DICARB within MCM).

2.) The effect of $HO_2$ heterogeneous uptake on the radical budget is likely over-stated. As the authors noted, the $HO_2$ uptake coefficient is highly uncertain. A relatively large value (i.e., 0.2) is chosen. In dense BB plumes, organic aerosol is the dominant composition with mass fraction up to 80%. As discussed in Abbatt et al. 2012 and George et al., the $HO_2$ uptake coefficient to solid organic particles is < 0.001 and to liquid organic particles is < 0.01. If applying the small gamma values, the $HO_2$ heterogeneous is negligible even in dense BB plumes.

**Response:** We stated in our initial text that we used a fairly large uptake coefficient, so we were immediately in some agreement with this comment. Since we have already varied our uptake coefficients used for several model simulations (now shown in Fig. 9 of SI), we added additional text to clarify this issue. We now state that model results acquired using an uptake coefficient of 0.02 leads to near negligible differences when compared to models that include no heterogeneous chemistry. For our statement where we extrapolate the impact of heterogeneous chemistry to more concentrated plumes (> 100 ug m$^{-3}$), we restate that we have used a fairly high uptake coefficient to underscore the uncertainty.

**Revised Text** The original paragraph focusing on heterogeneous chemistry has been mostly reworked into the new 'Model Sensitivity' Sect. 3.4 as its final paragraph (see full revised text above on page 2). This paragraph (lines 535-547) is nearly entirely revised. Included here is the most relevant text of this revised paragraph:

We focus our heterogeneous chemistry sensitivity tests on a 40 min period during the 17 August BB event in which OA concentrations were 30 μg m$^{-3}$. Inclusion of heterogeneous chemistry with standard parameter settings ($\gamma = 0.2$) leads to a 11% decrease in [$HO_2$], whereas use of a much higher and likely unrealistic $HO_2$ uptake coefficient of 0.5 resulted in a 25% $HO_2$ decrease. A similar $HO_2$ decrease results from using a higher specific surface area of 10 m$^2$ g$^{-1}$. Use of a smaller $HO_2$ uptake coefficient of 0.02 led to a nearly negligible decrease in [$HO_2$]. Constraining the OA concentration at 100 μg m$^{-3}$ for the same period - much higher than actually observed – leads to heterogeneous [$HO_2$] loss of 3% for an uptake coefficient of 0.02 and 30% for an uptake coefficient of 0.2. The only conditions in which heterogeneous loss of $HO_2$ to BB smoke would appear to be important would be for less dilute plumes ([OA]] > 100 μg m$^{-3}$) and a high uptake coefficient ($\gamma > 0.2$).

Text in the conclusion has been revised (line 573). The revised portion is highlighted.

Finally, the role of heterogeneous $HO_2$ uptake as a ROx sink would benefit from more direct measurements of particle size distribution and knowledge of $HO_2$ uptake coefficients. Heterogeneous $HO_2$ uptake was minimal for the dilute BB plumes studied here and would appear to only be important in less dilute BB plumes if the uptake coefficient is relatively high (e.g., 0.2).

3.) Missing HONO as model input is invoked in several places as a possible reason for the model vs measurement difference, but this reasoning is questionable. First of all, the delta_HONO/delta_CO after 3hr aging of BB plume, as shown in Peng et al., is 0.1 ppt/ppb, rather than 1 ppt/ppb as quoted in the manuscript (Line 377). In addition, Peng et al. clearly stated that after 3hr the [HONO] is near or below the instrument detection limit and they refrain from interpreting what it implies in terms of potential HONO steady state in aged plumes. Lastly, HONO can be

added to the box model (using delta_HONO/delta_CO = 0.1 ppt/ppb and measured CO), to directly test the effect of missing HONO on the modeling results.

**Response:** We agree that further investigation into HONO as a potential HOx source would improve this manuscript. We have updated the typo (1.0 pptv ppbv$^{-1}$ vs 0.1 pptv ppb$^{-1}$) that all reviewers identified. We have included additional model simulations where we constrain HONO using CO data and selected values of ΔHONO/ΔCO, following the suggestion of Anonymous Referees #1 and #2. Due to the large discrepancy between measured and modeled XO$_2$ during the 17 August BB event, we specifically investigate HONO as a missing HOx precursor for this event. Simulations were run using a variety of HONO constraints (a range in ΔHONO/ΔCO were used), and resulting XO$_2$ results for 17 August have been included in the SI in Fig. S14. We find a rather large ΔHONO/ΔCO of 3.0 pptv ppbv$^{-1}$ is required for a similar XO$_2$ enhancement as measured. Additional text was added within our 'Model Sensitivity' Sect. 3.4 and is included below. We also include text added to the SI regarding the new Fig. S14. Relevant text in the SI and conclusion have also been updated accordingly.

**Revised Text (Sect. 3.4, Lines 529-536):**

   As the 15 pptv increase in [XO$_2$] observed on 17 August is not captured by model simulations, we include HONO as an additional model constraint in additional GEOS-Chem simulations (see Sect. S8 of SI). Constrained HONO concentrations were determined by the product of selected ΔHONO/ΔCO values to the measured CO mixing ratios during BB periods. To achieve a similar ~15 pptv XO$_2$ enhancement as measured during the 17 August BB event, a HONO enhancement ratio of near 3 pptv ppbv$^{-1}$ is required, which provides an additional 0.15 - 0.60 ppbv HONO throughout the BB period. This ΔHONO/ΔCO value is 30 times larger than observed by Peng et al. (2020) for similarly aged plumes. While this value is likely unrealistic, larger ΔHONO/ΔCO have been reported by Peng et al. (2020). Other unmeasured ROx precursors were likely present and at least partially responsible for the elevated XO$_2$ concentrations observed.

**Additional SI Figure:**

[Figure]

**Figure S14.** F0AM $XO_2$ predictions with constrained HONO for 17 August period. Model results were obtained using the base version of GEOS-Chem. The initial HONO data constrained is based on the model outputs of HONO acquired using GEOS-Chem. Enhancements in HONO are then added for the 17 August BB event at 15:27 MDT using HONO enhancement ratios ($\Delta HONO/\Delta CO$).

Relevant text within SI for Fig. S15:

To investigate the importance of HONO in the model results, we ran additional F0AM simulations using the GEOS-Chem mechanism with HONO concentrations constrained using a range of $\Delta HONO/\Delta CO$ values and the measured CO mixing ratios. The raw enhancements in [HONO] for this event were 0.005 – 0.020 ppbv, 0.050 – 0.200 ppbv, and 0.150 – 0.600 ppbv when the $\Delta HONO/\Delta CO$ value was set as 0.1 pptv ppbv$^{-1}$, 1.0 pptv ppbv$^{-1}$, 3.0 pptv ppbv$^{-1}$, respectively. $XO_2$ predictions for the 17 August BB event are shown in Fig. S14. In order for the model to produce the 15 pptv measured increase in [$XO_2$] we need to constrain HONO during BB influenced period using a rather large $\Delta HONO/\Delta CO$ of 3.0 pptv ppbv$^{-1}$ – much greater than the $\Delta HONO/\Delta CO$ ratios observed for similarly aged plumes by Peng et al. (2020). While this value is likely unrealistic, larger $\Delta HONO/\Delta CO$ have been reported by Peng et al. (2020) and can be implied up to 5.9 pptv ppbv$^{-1}$, though for considerably younger smoke plumes, using temperate forest emission factors (Akagi et al., 2011).

**Additional Changes to Abstract and Conclusion:** Since the HONO enhancement ratio is rather large, we remove its mention from the abstract and replace with only 'unmeasured HOx sources'. Our statement regarding that 17 August event within the abstract has been updated:

One period of BB influence had distinct measured enhancements of 15 pptv $XO_2$ that were not reflected in the model output, likely due to the presence of unmeasured HOx sources.

Within the conclusion we now state the $\Delta HONO/\Delta CO$ value required for the 15 pptv rather than invoking its potential as a HOx precursor during the 17 August event (lines 565-566):

HONO photolysis likely contributed to the enhanced concentrations in [$XO_2$] measured during the 17 August BB period, though a rather large $\Delta HONO/\Delta CO$ value near 3.0 ppbv ppbv$^{-1}$ is necessary for a similar $XO_2$ enhancement.

4.) Figure 5 is interesting and related discussions should be expanded. For example, what does the lack of correlation between $P(O_3)$ and NO when $P(ROx)$ is small suggest. Does it suggest $O_3$ formation is VOC-limited? More information can be distilled from the figure, or from the four variables in the figure ($P(O_3)$, $P(ROx)$, NO, and HCN).

**Response:** We do not feel comfortable commenting on the lack of correlation in Fig. 5b ( $P(O_3)$ vs NO) due to the noise associated in our $P(O_3)$ calculation. At low $P(ROx)$ values, $P(Ox)$ is low - typically less than 2 ppb hr$^{-1}$. The large variability in $P(Ox)$ for this subset of data precludes making any conclusions regarding its sensitivity to [NOx].

5.) In the manuscript, some metrics are from direct measurement (e.g., [XO2]), some metrics are calculated from measures species (e.g., $P(Ox)$ from Eqn. (3)), and some metrics are purely based on box model. These need to

carefully worded in the discussion to avoid confusion. For example, Line 460 and others use the term "measured OH reactivity", which the reader believe is the calculated OH reactivity based on measured VOCs. Then, Line 477 refers to Eqn. (3) as measured P(ROx), which makes the reader confused for a bit, as it is calculated, not measured. Such subtleties hinder the readability of the manuscript.

**Response:** We agree with this issue regarding the clarity of wording for metrics calculated from direct measurements. This has been addressed throughout the paper. There are two instances in which "measured OH reactivity" was mentioned (now lines 473 and 780). This wording has been changed to "OH reactivity attributed to measured compounds". The term "measured P(ROx)" has been changed to 'measurement-based P(ROx)'.

6.) Line 482. Are there any constraints or independent verification on the contribution of photolysis of methyl glyoxal and glycolaldehyde to the P(ROX)? Their contributions seem much larger than expected, based on measurements of these two species in previous wildfire studies.

**Response:** The contributions of methyl glyoxal was initially understated, and the text has been updated to state that the methyl glyoxal contributes 26% (previously listed as 19%) to the dominant 'carbonyl + hv' P(ROx) category. Further discussion focusing on the significant carbonyls contributing to this P(ROx) category has been added. We focus on methyl glyoxal, glycolaldehyde, and a dicarbonyl formed from BB VOC chemistry. We have included text stating that the precursors to methyl glyoxal and glycolaldehyde are from the constrained compounds of MVK/methacrolein and 2-MBO, respectively. We add context to this issue by listing concentrations of important species, and for methyl glyoxal we compare our modeled concentrations to values measured by other studies.

**Revised Text:**

Relevant text within our model-based P(ROx) discussion (lines 493-501) is provided with newly added lines highlighted.

Photolysis of carbonyls accounts for most of the remaining modeled daytime P(ROx), though only a fraction (less than 15%) of this category is from the measured carbonyl compounds acetaldehyde and acetone. Unmeasured carbonyls account for the rest, with methylglyoxal and glycolaldehyde accounting for 26% and 8%, respectively. Predicted methylglyoxal concentrations were typically between 0.4 and 0.6 ppbv. These concentrations are about an order of magnitude greater than those measured at mountaintop sites (Mitsuishi et al., 2018;Kawamura et al., 2013) but lower than those observed at a suburban site in China (Liu et al., 2020) and in biomass burning plumes observed in the Amazon (Kluge et al., 2020). Methylglyoxal is largely formed from the oxidation of MVK and MACR, themselves oxidation products of isoprene. A BB VOC related dicarbonyl, 4-oxo-2-pentenal (listed as C5DICARB within MCM), which is formed from methyl furan oxidation, accounted for 13% of P(ROx) from carbonyl photoloysis.

**Minor comments from Anonymous Referee #1:**
Line 26-28. The reader suggests to rephrase the sentence to "the model over-estimates the $XO_2$ by 30%".

**Response:** We have revised the abstract. The phrasing of "the model agreed within 31%" has been changed to more closely reflect this suggestion.

**Revised abstract text (Lines 28-29):**

The models consistently over-estimated $XO_2$ with the base MCM and GEOS-Chem $XO_2$ predictions high by an average of 28% and 20%, respectively.

Line 29-30. Suggest rephrasing to "likely due to the presence of an unmeasured HOx source that is not included in models". The whole sentence may need to be rewritten after investigating the role of HONO as mentioned above.

**Response:** Since we have found that rather a large HONO enhancement ratio was required for modeling a similar $XO_2$, we have updated the sentence to more closely resemble the above suggestion.

**Revised Abstract Text (Lines 29-30):**

One period of BB influence had distinct measured enhancements of 15 pptv $XO_2$ that were not reflected in the model output, likely due to the presence of unmeasured HOx sources.

Figure 5b. $P(O_3)$ is used in the y-axis, but $P(Ox)$ is used in the figure caption. Be consistent.

**Response:** $P(O_3)$ within the figure and any mentions in the text related to Figure 5b (lines 499-502) have been updated to $P(Ox)$. A similar comment from Anonymous Reviewer #2 suggested introducing a legend to figure 5a. The revised figure is presented here.

**Revised Figure 5:** Figure 5a with an included legend. Figure 5b with '$P(Ox)$' replacing '$P(O_3)$' of y-axis title.

[Figure]

a.

b.

Line 387-397. The negative delta ΔO₃/ΔCO on 8/16 is likely a result of inaccurate O₃ background and mixing, rather than depletion. It is because as the authors noted, the [NO₂] is much smaller than [O₃], suggesting that [Ox] ~= [O₃], which rules out the possibility of depletion.

**Response:** This section (lines 399-409) has been rewritten in response to Anonymous Referee #3, but we have retained the -0.02 ΔO₃/ΔCO value for 16 August. This ΔO₃/ΔCO value was determined by subtracting background [O₃] and [CO] from plume values. Background O₃ and CO concentrations were stable. Plume concentrations (red shaded period of 16 August on period 3) also remained stable for both species for about 1 hr. Depletion of this magnitude is common for young plumes aged less than 1 day (see ΔO₃/ΔCO table within the Jaffe and Wigder, 2012 review).

Be consistent with which model results are presented. For example, why is MCM-BBVOC-het presented in figure 8, but MCM-BBVOC in figure 6?

**Response:** Model results within the figures are consistently shared for MCM-BBVOC (without the additional heterogeneous chemistry) throughout the manuscript (Fig. 5 and Fig. 6). We only present MCM-BBVOC-het results in Fig. 8 to show its contribution to L(ROx).

**Report #2 by Anonymous Referee #2**

First, the authors find that the model was not able to reproduce the $XO_2$ enhancement of 15 pptv during the Aug 17 smoke event, and suggested that HONO, which was not measured, may be the cause. This possibility is not really investigated in the paper. What magnitude of HONO would be needed to achieve the necessary production of $XO_2$? It would be useful for the authors to verify their assumptions by conducting sensitivity analysis of the modeled HONO (perhaps by applying a scale factor to CO or $NO_2$).

Response: A similar comment was made by Anonymous Reviewer #1. See the response to comment 3 on page 4 of this document. Simply put, additional model simulations were conducted using HONO concentrations that were constrained using a variety of $\Delta HONO/\Delta CO$ (ranging 0.1 up to 3.0 pptv ppbv$^{-1}$). We share that a $\Delta HONO/\Delta CO$ value near the 3.0 pptv ppbv$^{-1}$ value is required for a similar enhancement measured. As this is a rather large $\Delta HONO/\Delta CO$ value for a several hour aged plume, we suggest that the $[XO_2]$ enhancement was likely caused by several unmeasured HOx precursors.

Second, there is an inadequate amount of discussion of how the modeled radical precursors compare to measurements. The authors showed significant model overestimates in $XO_2$, $P(ROx)$, and $P(O_3)$ across all model mechanisms used, but did not discuss in detail the possible causes of model discrepancies. More discussion on the cause of the model overprediction and what might be pursued to reconcile the problem would be useful. Are there inaccuracies in the recycling within $XO_2$ species? What does the diurnal profile of model-obs discrepancy look like? What possible mechanisms could bridge the gap between measured and modelled total $XO_2$? Is the modeled NOx close to observation? How is NOx constrained in the model (e.g., kept $NO/NO_2$ ratio the same as observation or not)? This kind of analysis may help better understand where the model-measurement discrepancy arises from.

Response: We agree that investigations into possible causes for the discrepancy between the measurements and models are useful. Additional model simulations have been conducted focusing on model sensitivity to a variety of parameters. We now include an additional 'Model Sensitivity' section (Sect 3.4), within the 'Results and Discussion'. The discussion of model results has been restructured to accommodate the new content and improve readability. See the page 1 response to the first Anonymous Referee #1 comment which includes the full text of Sect 3.4.

Regarding the species constrained within the model, NO and $NO_2$ were constrained individually – this was stated originally in the SI but additional text has been added to Sect. 2.6 'Zero-Dimensional Modeling' for clarity. Therefore, there is no model-measurement discrepancy for NOx. We have also added a list in the main text of some notable compounds constrained. Newly added Sect. 2.6 content regarding model set up is highlighted in the revised text below.

Potential causes of the model overprediction and discrepancies are introduced in Sect. 3.3 'Model Evaluation' with the discussion surrounding the large contributions of unmeasured carbonyls to P(ROx) and unmeasured compounds to OH reactivity. The new 'Model Sensitivity' section focuses on how parameters impact as model-measurement discrepancies in $XO_2$. As there is a large buildup of secondary compounds, we investigate model sensitivity to the first order dilution applied to non-compounds (see time series outputs from different dilution settings in new SI Fig. S15 shown on page 3). Simulations with HONO constraints were conducted and are discussed as a potential HOx source for the 17 August event where models lacked the 15 pptv enhancement observed. We have also moved much of and expanded our heterogeneous chemistry discussion to this new 'Model sensitivity' section (see response to comment 2 by Referee #1 on page 4).

**Revised Text:**

The new Sect. 3.4 'Model Sensitivity' text is included in response to Anonymous Referee #1. See page 1.

**Sect. 2.6 Zero-Dimensional Modeling Revision (Lines 303-334):**

We include this entire section here with newly added content highlighted.

[revised manuscript text omitted]

Last, the ozone analysis could be better utilized to reconcile the calculated rate of production with measurements. For example, the authors could show the changes in measured-to-modelled ratio of $P(O_3)$ and $XO_2$ with different parameters (e.g., NOx concentration), colored by different BB events, to find plausible explanations of the discrepancy.

**Response:** Our response to the previous comment discusses additional model simulations conducted to investigate overall discrepancies between the models and measured values. That response included model sensitivity to changes in parameters such as NOx, which is suggested by this comment. Therefore, the previous response mostly covers this comment. The analysis suggested in this comment focuses on discrepancies during the BB periods. However, there were constant discrepancies during all daytime periods. The only clear additional discrepancy modeled during a BB period was the 17 August plume where the ~15 pptv $XO_2$ enhancement was measured only. For that discrepancy, we suggested the presence of unmeasured HOx precursors including HONO as stated in the first response of this section.

**Specific comments:**

1. How does the wind speed, RH, temperature vary each day? A brief description of meteorological conditions would be helpful. Or you could add a panel for that in Fig 2.

    **Response:** We agree that these meteorological conditions can be useful for a variety of reasons. We have included a brief mention of temperature ranges in the main text and have included a figure within the SI.

    **Revised Text (Lines 204-213):**

    Meteorological measurements were made both on the AML and permanently at the McCall site. Temperature, wind speed, and wind direction were collected by a 3-D R.M. Young (Model 81000RE) sonic anemometer stationed permanently at the McCall site at a height of 10 m. Additional wind was measured with a 2D R.M. Young (Model 81000RE) sonic anemometer mounted to the AML rooftop and corrected for speed and truck orientation with data from a Hemisphere GPS compass (model Vector V103). Temperature, RH, and wind data are shared in the supporting information (Fig. S3). Daily maximum temperatures ranged 22 ℃ and 28 ℃ while minimum temperatures ranged 4 ℃ to 13 ℃. Solar irradiance was measured by a permanently stationed ARISense air quality sensor system (Cross et al., 2017). This was used to derive photolysis frequencies of interest, such as $J_{NO_2}$, by scaling measured irradiance to outputs from the National Center for Atmospheric Research (NCAR) Tropospheric Ultraviolet and Visible (TUV) radiation model. This process for deriving photolysis frequencies is described in greater detail in the supplement.

    **Figure S3 Added to SI**

[Figure]

**Figure S3.** Meteorological data for the FIREX 2018 campaign. Periods of smoke are shaded as per Fig. 2 of the main text.

2. It would be more clear if you move part of the key model description from the SI (Line S108-114) to the main text, mainly what and how the measurements were used to constrain the model and what values are used for unmeasured species.

    **Response:** We agree that including more of the key modeling content would clear things up for the reader. We have now stated that all available measurements were used to constrain the model, and we have listed some of the more important constrained species to the main text but have kept the full list in the SI. All unmeasured species were set with initial 0 ppbv concentrations as additional simulations using a spin up period had minimal effects on our model output of interest of $XO_2$. The background concentrations set for unmeasured species is now stated.

    **Revised Text:** Revised text is listed in response to a prior comment. See the updated Sect. 2.6 'Zero-Dimensional Modeling' text on page 11. New text focused on model set up is highlighted.

3. Line 376-377: Peng et al. showed HONO enhancement ratios decay to ~ 0.1 ppt/ppb instead of 1 ppt/ppb for aged (> 3h) plumes. What enhancement ratio is needed to resolve the model disagreement if you initialize the model with HONO that scales with CO (or NOx)?

    **Response:** The quoted Peng et al. HONO enhancement ratio has been updated to 0.1 pptv ppbv[-1]. We have included additional simulations where HONO is constrained using CO data and selected $\Delta HONO/\Delta CO$ values. $\Delta HONO/\Delta CO$ near 3 pptv ppbv[-1] is required.

See the response to comment 3 from Anonymous Referee #1 (page 4) for greater detail and any revised text.

4. In line with the previous point, given the importance of HONO as an OH precursor, how does PO3 change after including HONO in the model? And how does that affect NOx? Is the estimation of the missing HONO different across different BB events?

   **Response:** Given that we constrained NO and $NO_2$ individually, NOx is not affected by any HONO constraints. Since a missing HOx source is not as clearly evident within the other four plumes, we only focus our analysis of HONO for this one plume. P(Ox) increases by ~10% with HONO constrained using a 3.0 pptv ppbv$^{-1}$ enhancement ratio during that 17 August episode.

5. Figure 8: Why is the modeled P(ROx) almost a factor of 2 larger than observed? Line 481-482 suggests that the greatest HOx contributors in the carbonyls group are unmeasured carbonyls - how are they initialized in the model? How does each modeled radical precursor compare to the observation? Where are the sources of discrepancy in P(ROx)? It is important to mention how the uncertainty of the carbonyls input could affect the results and conclusions.

   **Response:** A relevant response is given to a comment focused on the verification of notable carbonyls in the model (see page 7, 'specific comment 6' of Anonymous Referee #1). There, additional text focused on P(ROx) is included. In our initial P(ROx) discussion, we did not mention that the overprediction of modeled P(ROx) is for the latter half of the campaign was lower and is within ~60% of daytime measurement-based P(ROx) (see revised text below). The unmeasured carbonyls are not initialized in the model, and we add context to their sources in our related page 7 response. Common modeled-measured radical precursors are the same due to each measured precursor being explicitly constrained, so some model overprediction in P(ROx) is absolutely expected.

   **Revised Text (Line 512):**
   Results from the 21 August through 24 August period (see Fig. S17 in SI) are similar to the results presented above for 16 – 18 August, though the unmeasured portion of P(ROx) is smaller for the former. This resulted from the considerably smaller concentrations of MVK, MACR, isoprene and BB VOCs measured during this period. The smaller portions for the carbonyls and 'alcohols, acids' categories led to smaller P(ROx) totals that peaked near 0.7 pptv s$^{-1}$ rather than the calculated 1.2 pptv s$^{-1}$ for the period shown in Fig. 8.

6. Do $XO_2$ measurements agree with the $XO_2$ estimates derived from modified Leighton ratio calculated with measured ozone, NO and $NO_2$ concentrations from the campaign? A scatter plot of derived $XO_2$ versus ECHAMP observations could be made.

   **Response:** While $XO_2$ estimated by the Leighton ratio (assuming photostationary state) can be useful, it is extremely sensitive to even small deviations from photostationary state (common during partially

cloudy days) and to the uncertainties in the measured values ($J_{NO2}$, NO, $NO_2$, and $O_3$) and the relevant rate constants. For typical daytime values at McCall, the combined uncertainty in $XO_2$ values calculated assuming photostationary state are usually greater than 100%. For this reason we have chosen not to include this line of analysis.

**Minor Comments:**

1. Fig 2: How is smoke presence determined? There seems to be no text descriptions.

   **Response:** Within Fig 2., we shade red for significant smoke ([HCN] > 1 ppbv) and tan for general smoke presence. There was no clear in text description for general smoke presence, so we have included an additional sentence for clarity. The general 'smoke presence' was identified using all Fig. 2 smoke tracers (HCN, $CH_3CN$, organic PM, and CO) for periods before and after the distinct BB plumes.

   **Revised Text (Line 227):**

   General smoke presence (tan shaded regions in Fig. 2) was identified before and after each significant smoke period when background smoke tracer concentrations remained elevated compared to stable background air.

2. Line 186: extra period and space

   **Response:** This extra period and double space has been removed.

3. Line 187: missing closing parenthesis

   **Revised Text (Line 190):**

   VOC measurements were made by an ARI Vocus (proton-transfer-reaction high-resolution time-of-flight (PTR-HR-ToF) mass spectrometer) (Krechmer et al., 2018).

4. Line 302, Extra space after "MCM-base"

   **Response:** This extra space has been fixed.

5. Line 357, sentence incomplete: "Clusters of data points at HCN concentrations ARE…."

   **Revised text (Line 369):** This has been addressed.

   Clusters of data points at HCN concentrations below 0.75 ppbv are observed for the 18 August and 24 August data sets.

6. Figure 3: better have a legend for the bottom panel

   **Revised Figure:**

   Fig. 3 with the inclusion of a P(ROx) legend

[Figure]

7. Figure 5. Add legend for model vs obs in panel a for convenience of the readers; also specify the analysis is for Aug 17 event in the caption

**Response:** A legend has been added to the diurnal P(Ox) plot as requested. The caption has not been revised because this analysis includes all campaign data. As a suggestion to retitle the y-axis from Anonymous Referee #1, the revised figure that now includes a legend is shown on page.

8. Fig 6. Typo in the legend for the upper panel: ISOPO2 instead of Isoprene?

**Response:** The 'isoprene' $XO_2$ category includes ISOPO2 along with additional $RO_2$ species produced from isoprene oxidation. This was mentioned in the text (line 453) prior to revisions: "Modeled $XO_2$ comprises $HO_2$ (typically ~45-50%), $CH_3O_2$ (~20-25%), and the remaining portion a combination of $CH_3CO_3$, **RO₂ derived from isoprene oxidation**, and other organic peroxy radicals." We have updated the caption to Fig. 6 to avoid further confusion.

**Revised Fig. 6 Caption (newly added text is highlighted):**

Time series of modeled OH, $HO_2$, speciated $RO_2$, and OH reactivity (OHR). These results were acquired using the MCM-BBVOC mechanism. Periods of smoke are shaded as per Fig. 2. Measured $XO_2$ is included as black markers for comparison. Legend categories for $XO_2$ and OHR are mostly straightforward. Note that the isoprene $XO_2$ category includes several $RO_2$ species produced from isoprene oxidation.

9.  Line 471-472: Did the decrease in OHR happen upon smoke arrival? From Fig 6 it seems to be upon smoke exit?

    **Response:** The OHR decrease happened upon smoke departure. The text has been revised.

    **Revised Text (Lines 566-590):**

    Changes in OH reactivity were observed for some smoke periods. Subtle changes were noted during the 16 and 17 August events due to changes in the inorganic, carbonyl, biogenic, and BB VOC portions. A noticeable decrease in reactivity from 10 to <5 s$^{-1}$occurred on 24 August upon smoke departure. Decreases in nearly all reactivity categories contributed with notable contributions due to depletion in aldehyde concentrations and [CO]. This period sustained the lowest reactivities for the entire data set while also having the lowest concentrations of smoke tracers.

10. Line 477-478: shall be "O(1D) reaction with H$_2$O"

    **Response:** This typo has been fixed. This initially read, "O(1D) reaction with O$_3$".

11. SI Line 151: fix typo "GOES-Chem"

    **Response:** This typo has been fixed to "GEOS-Chem".

12. Fig S10 and 11: fix caption placeholder (Fig X)

    **Response:** These placeholders have been updated to correctly reference the previous SI figure.

**Point-by-point Response to Reviewer's Comments**

**Report #3 by Anonymous Referee #3**

**Major Comments:**

Line 387 etc: I understand that the dO$_3$/dCO metric has been used before, but it seems to me that it is undercontrained here. Wouldn't the derived slope be strongly dependent on the time of day when the measurements were taken? O$_3$ has a strong diurnal cycle due to photochemistry, while CO does not. So across the several hours of measurements encompassing in-plume as well as pre/post plume background, the dCO term will likely be dominated by the addition of CO in vs out of the plume. However, the dO$_3$ term will be a sum of the O$_3$ added (or lost) from plume-specific chemistry plus any O$_3$ that would have been added (or lost) during the several hour measurement period due to regular photochemistry. With this in consideration, it appears that the slopes in Fig. S6 are most positive for the 22$^{nd}$ and 23$^{rd}$ because those data were taken during the early part of the day when ambient O$_3$ concentrations were increasing most rapidly, while the slope for the 17$^{th}$ was smaller because the data came just shortly before peak daily O$_3$, and the slopes for the 24$^{th}$ and 16$^{th}$ (not shown) were smallest because the data came from after the daily peak of O$_3$ when mixing ratios were declining slowly. This might correlate with your calculated P(Ox), though it would depend on L(Ox) too I think. In other words, this method could work if it was using dO$_3$/dCO data that came from a short enough time span that the daily O$_3$ cycle was not a factor, for instance aircraft measurements spanning just a few minutes. But here, you're deriving a slope between in-plume data and out-of-plume data that are separated in time by several hours, during which time the actual O3 in each airmass is changing (and changing in different ways depending on time of day of sampling). Therefore I do not think you can use this analysis to support your abstract-level conclusion that "During BB events, O3 concentrations were enhanced…" at line 23. That conclusion may still be accurate, but you will need to provide better evidence.

**Response:** We agree that the $\Delta$O$_3$/$\Delta$CO values we initially presented are indeed impacted by the time of day and not solely due to the impact of smoke photochemistry. The 22 August and 23 August $\Delta$O$_3$/$\Delta$CO values are significantly affected by the increase in O$_3$ observed during that time of day regardless of the presence of smoke. This led to the high $\Delta$O$_3$/$\Delta$CO values. However, we feel confident that the values we present for the smoke events of 16, 17, and 24 August can be attributed to smoke photochemistry. While the 22 August and 23 August smoke events occurred in the early afternoon with subtle changes in O$_3$, the other 3 events all occurred after 15:00 MDT and had distinct and more immediate changes in O$_3$. Furthermore, these events had stable levels of [O$_3$] and [CO] during both smoke and background periods. In response to this comment, we omit the $\Delta$O$_3$/$\Delta$CO analysis for the 22 August and 23 August events, and we continue to present $\Delta$O$_3$/$\Delta$CO for the other BB events of 16, 17, and 24 August. The text within both 'Calculations' section (Sect. 2.5) and 'Ozone Production' section (Sect. 3.2) have been updated to address these changes. Data for 22 and 23 August has been omitted from Fig. S8 (previously Fig.

S6) to reflect these changes. Within the abstract, the range in $\Delta O_3/\Delta CO$ is reported. The range in the abstract has been updated accordingly.

**Revised Text** (revised portions related to this comment have been revised)**:**

Lines 271-281 (Sect. 2.5 'Calculations' within Sect. 2 'Methods')

Ozone enhancement ratios $\Delta O_3/\Delta CO$ from smoke influence were determined for the most distinct BB events of 16, 17, and 24 August. For the 17 and 24 August events, the ozone enhancement ratio was determined using the York bi-variate linear regression method (York et al., 2004) using a continuous section of $O_3$ and CO data that includes 60 minutes of background air, a transitional smoke period (tan shaded regions in Fig. 2), and 60 minutes of significant smoke period data (red shaded regions in Fig. 2). Enhancements in $NO_2$ were typically under 0.2 ppbv and so the difference between considering $\Delta Ox$ ($Ox = O_3 + NO_2$) and $\Delta O_3$ was negligible. The linear regressions are included in the supplement (Fig. S8). $\Delta O_3/\Delta CO$ for the 16 August event was determined using Eq (1) with $O_3$ and CO data collected during a stable period at the start of the significant smoke period and a stable background prior to smoke presence.

$$\Delta O_3/\Delta CO = ([O_3]_{Smoke} - [O_3]_{Background})/([CO]_{Smoke} - [CO]_{Background}) \qquad (1)$$

This event had a temporary depletion in $O_3$ by ~20 ppbv for the start of smoke significance, then returned to near background levels of $O_3$. $\Delta O_3/\Delta CO$ values were not calculated for the remaining 22 August and 23 August smoke events. These events had less distinct $O_3$ enhancements and occurred at times that $O_3$ increased during non-smoky time periods.

Lines 399-409 (Sect. 2.5 'Calculations' within Sect. 3 'Results and Discussion')

We describe the extent of ozone formation for the 16 August, 17 August and 24 August BB influenced periods using the commonly used $\Delta O_3/\Delta CO$ metric. These values depict the $O_3$ produced in transit to the McCall site while accounting for plume dilution or overall smoke influence of the site air sampled. $\Delta O_3/\Delta CO$ values were -0.02, 0.06 and 0.03 ppb ppbv$^{-1}$ for the 16, 17 and 24 August smoke events, respectively. These calculated values fall within the wide variability and range of literature $\Delta O_3/\Delta CO$ values for boreal and temperate forest fire smoke plumes aged less than two days, including numerous examples of ozone depletion for aged plumes (Jaffe and Wigder, 2012). Though the smoke was likely sourced from the same wildfire for the 16 August and 17 August events (section 2.4), we observe $O_3$ depletion on 16 August and $O_3$ enhancement on 17 August. $\Delta O_3/\Delta CO$ values were not calculated for the 22 August and 23 August smoke events as we were unable to attribute the observed increases of $O_3$ to smoke influence as they occurred at the same time that $O_3$ usually increased during non-smoky time periods as mentioned in the section 2.5. Ox enhancement ratios are not presented but differed insignificantly from $O_3$ enhancement ratios as $NO_2$ concentrations were much lower than $O_3$ concentrations (see Fig. 3).

**Revised Figure (Fig. S8 in supplement):**

[Figure]

**Figure S8.** Comparisons of 2-minute $O_3$ and CO data that were used to derive biomass burning ozone enhancement ratios ($\Delta O_3/\Delta CO$) for the 17 and 24 August BB smoke events. Each set of data plotted was collected continuously and with the selected periods including 60 minutes of background air, a transition to or from BB smoke influence, and 60 minutes of significant BB smoke influence ([HCN]>1.0 ppb).

Fig. 4: The same analysis as my previous comment applies here; $O_3$ has a prominent diurnal cycle that would exist with or without a wildfire plume, while HCN would not. Plume chemistry may enhance the $O_3$ cycle, but it apparently doesn't dominate it. Thus, the trends shown in this plot are driven by that underlying $O_3$ diurnal cycle. The slopes are steep and positive for data taken when a plume arrived during midmorning when $O_3$ is increasing due to ambient photochemistry, and shallow for data taken when a plume arrived after peak daily O3. If you had a plume arrive at ~midnight, $O_3$ would be decreasing while HCN increased, giving a negative slope in Fig. 4. Therefore I do not think you are correctly interpreting the apparent positive correlation between $O_3$ and HCN here. The proper correlation would be between only the $O_3$ derived from smoke plume chemistry (i.e. subtracting the normal background photochemical diurnal cycle) vs. HCN. Obviously that's really difficult to parse and would likely require a model, so I'm not sure what to suggest other than removing this part of the analysis.

**Response:** We simply use Fig. 4 ($O_3$ vs HCN) to show that the greatest [$O_3$] occurred during the smokiest periods (highest HCN values) and do not ascribe any particular meaning to the $O_3$/HCN ratio. The negative slope comment for regarding any possible nighttime HCN enhancements is irrelevant because Fig. 4 only includes daytime data

(9:00 to 22:00 MDT) as stated in both the text and figure caption. Therefore, we have chosen to keep this figure and respectfully disagree that it should be removed.

Line 354: Also, can you state here what is the R^2 value for the correlation in Fig. 4?

**Response:** As per the response to the previous comment, we are not presenting an $O_3$/HCN slope and merely include this plot to point out that the highest $[O_3]$ values occurred during the smokiest periods. Since we are not presenting any slopes, we think that it would be misleading to include an $R^2$ value as it would suggest that the correlation between $O_3$ and HCN should be considered overly quantitatively.

Line 376: I believe j(HONO) was measured, or could be estimated. Can you do a rough calculation of how much [HONO] would be needed to account for the added $XO_2$? That would be helpful for convincing the reader that it is reasonable to assume HONO is the missing factor.

**Response:** Similar comments have been made by Anonymous Referees #1 and #2. See our response to Anonymous Referee #1 at the end of page 4. Rather than doing a rough calculation to suggest the HONO enhancement required to achieve a 15 pptv enhancement in $XO_2$, we conducted additional model simulations where a range in HONO concentrations were constrained by applying different $\Delta HONO/\Delta CO$ values. An enhancement of up to 0.6 ppbv HONO is required suggesting a $\Delta HONO/\Delta CO$ value of near 3.0 pptv ppbv$^{-1}$.

Line 377: The fit line from Peng et al. 2020 (Fig. 3) suggests a value of more like 0.1 pptv/ppbv$^{-1}$ for average dHONO/dCO at 3 hours age, not 1 pptv/ppbv$^{-1}$ as stated here. However, the line is fit to all data regardless of in situ j$_{HONO}$, and some plumes did have elevated [HONO] at those ages, so it's possible this plume also did.

**Response:** The incorrect $\Delta HONO/\Delta CO$ value of 1.0 pptv ppbv$^{-1}$ has been corrected to the correct value of 0.1 pptv ppbv$^{-1}$. We also state that Peng et al. (2020) observed a range of values (line 531): "This $\Delta HONO/\Delta CO$ value is 30 times larger than observed by Peng et al. (2020) for similarly aged plumes. While this value is likely unrealistic, larger $\Delta HONO/\Delta CO$ have been reported by Peng et al. (2020)."

Line 404: Here you say "…there is little correlation between P(Ox) and smoke tracers." Then in the next sentences you discuss examples of how BB influences P(Ox). I would agree with the sentence at line 404. Also, Fig. 5a is not the correct plot to show to conclude that "P(Ox) is slightly higher during the afternoon and evening smoke influenced periods compared to non-smoke periods" as you say at line 408. I see the high [HCN] data covering a similar span of P(Ox) as low [HCN] data. Really to draw this conclusion you would need to split the green trace

(diurnal cycle of measured P(Ox)) into two, one with high HCN and one with low HCN to show the difference between them. Please do that, and then alter the text to tell a consistent story that there either is or is not a strong effect of smoke on P(Ox) in your measurements.

**Response:** We agree that the original text was not written clearly. As requested, we have created a smoke P(Ox) median line ([HCN] ≥ 1ppbv) and compared to a non-smoke median line (([HCN] <1ppbv). We do not include these results in the manuscript, but overall we have found negligible difference between P(Ox) for smoke and non-smoke data. The median diurnal cycles for smoke and non-smoke data are about the same at all times with the difference between smoke and non-smoke exchanging P(Ox) values alternating between negative and positive values. In response to these findings, we have slightly altered the text and have removed the previous statement that P(Ox) was greater in the afternoon and evening.

**Revised Text:**

Instantaneous $O_3$ production rates are calculated using NO and $XO_2$ concentrations (Fig. 3). Gaps in P(Ox) are due to measurement gaps in $XO_2$ when ECHAMP was offline for calibrations and diagnostic tests. The highest P(Ox) values occurred on 17 and 18 August during non-smoky periods between 10:00 MDT and 12:00 MDT, reaching formation rates slightly greater than 8 ppbv hr$^{-1}$. For the entire campaign, median P(Ox) peaked at 11:00 MDT at 5.8 ppbv hr$^{-1}$. As NO concentrations were low and rarely exceeded 1 ppbv, changes in [NO] had a near-linear impact on P(Ox). Noisy P(Ox) periods, such as the entire afternoon of 16 August, are mainly attributed to the atmospheric variability of and measurement precision for NO. Overall, there is little correlation between P(Ox) and smoke tracers. However, elevated P(Ox) during the 17 August event is somewhat evident. The ~27% increase in $XO_2$ and near constant value for NO led to this temporary increase in P(Ox). P(Ox) increased from ~2.5 to 8.9 ppb hr$^{-1}$ during the transition from background air to significant smoke, remained elevated for 34 minutes, and then returned to near background P(Ox) rates. The overall lack of impact of BB influence on P(Ox) is further depicted in the P(Ox) diurnal cycle of Fig. 5.  Modeled P(Ox) results for the same time period are also presented in Fig. 5a with the green median trend. These model results were acquired using F0AM with the MCM-BBVOC mechanism. Modeled P(Ox) is consistently greater than measured values with the greatest discrepancy occurring in the 7:45 to 8:15 MDT period. This difference is due to modeled [$XO_2$] being greater than measured [$XO_2$]. While we present results acquired using four unique chemical mechanisms, model predicted P(Ox) was always greater than measurements, though within the combined uncertainties.

Line 477: By plotting them together in Fig. 8, you're directly comparing measured P(ROx) from a subset of sources with modeled P(ROx) from all sources. They don't agree very well, nor should they agree, especially at night when alkenes+O3 dominates. But this just adds confusion, and it doesn't help answer the real question of how well does the P(ROx) from measured sources compare with the P(ROx) modeled from the same subset of sources? I'd suggest maybe adding a dashed line for the P(ROx) modeled from that same subset for comparison, or altering the figure in some other way to help clarify so you're not comparing apples and oranges.

**Response:** The model was constrained using all available measurements. Therefore, the modeled P(ROx) is based on our measurements but also includes additional ROx precursors that the model predicts (i.e., P(ROx) from measured sources and P(ROx) modeled from the same subset of sources are exactly the same, because it is constrained that way). The model categories of 'O($^1$D) + H$_2$O' and 'HCHO + hv' are strictly based on measurements and the carbonyl photolysis section includes photolysis of our constrained acetaldehyde and acetone measurements. The inclusion of measurement-based P(ROx) is there only to show the additional P(ROx) that is found by the model, largely due to the photolysis of unmeasured carbonyl compounds.

**Minor Comments:**

Line 33: add a citation here for Brazil and Australia fires

**Revised text (Line 33):** A citation was added for each fire.

For example, Brazil's Amazon rainforest wildfires in 2019 (Cardil et al., 2020) and Australia's bush fires in 2019-2020 (Yu et al., 2020) were both marked by historically high amounts of land burned.

Line 272: It would be helpful if you refer to a specific location in the SI, e.g. Sect. S4 here. Please check other references to the SI as well.

**Response:** This is a good suggestion. We have updated each mention of the SI so that we refer to a specific section or figure.

Fig. 5: The figure panels are labeled P(O$_3$), but should probably be P(O$_x$) to be consistent with the caption and rest of text (and be consistent throughout).

**Response:** The P(O$_3$) appearances in the text and Fig. 5 have been updated accordingly. See our page 8 response to Anonymous Referee #1 that includes the revised figure.

Line 384: I'd prefer if you labeled the green and blue traces using a legend in the figure, rather than having to dig through the caption to find that information. Same goes for the P(ROx) speciation in the bottom panel of Fig. 3.

**Response:** Anonymous Referee #2 has also requested figure legends. We present revised Fig. 3 and Fig. 5 on page 16 and 8, respectively.

Line 418: Model Evaluation should be Sect. 3.3, not 3.2.

**Response:** The 'Model Evaluation' section number has been updated accordingly. The numbers of sections and figures have been double checked to avoid any repeats or skips.

**Community comments**

**'Interpretation of 2B UV absorption O₃ measurement in wildland fire plumes' by Andrew Whitehill**

The manuscript claims that "$O_3$ was measured by a 2B-Tech UV absorption instrument." This instrument (along with most UV absorption instruments) measure ozone at 253 nm. UV photometric ozone monitors, including the 2B instruments, are known to produce significant positive interferences due to VOC's and particulate matter in wildland fire plumes. These interfrences tend to be correlated with CO concentrations. The Authors do not sufficiently address this analytical artifact in the manuscript and how they corrected for or addressed it. Given that the delta $O_3$ vs delta CO ratio was a critical aspect of this paper, authors should provide additional description of how they corrected their $O_3$ measurements for fire related smoke artifacts or provide a justification for why such an artifact is not present in their data.

References to VOC and particulate artifacts in UV-photometric ozone measurements:
Huntzicker, J. J. and Johnson, R. L., Investigation of an ambient interference in the measurement of ozone by ultraviolet absorption photometry, Environ. Sci. Tech., 13, 1414–1416, 1979.
Grosjean, D. and Harrison, J.: Response of chemiluminescence $NO_x$ analyzers and ultraviolet ozone analyzers to organic air pollutants, Environ. Sci. Tech., 19, 862–865, 1985.
Dunlea, E. J., Herndon, S. C., Nelson, D. D., Volkamer, R. M., Lamb, B. K., Allwine, E. J., Grutter, M., Ramos Villegas, C. R., Marquez, C., Blanco, S., Cardenas, B., Kolb, C. E., Molina, L. T., and Molina, M. J.: Technical note: Evaluation of standard ultraviolet absorption ozone monitors in a polluted urban environment, Atmos. Chem. Phys., 6, 3163–3180, https://doi.org/10.5194/acp-6-3163-2006, 2006.
Spicer, C. W., Joseph, D. W., and Ollison, W. M.: A re-examination of ambient air ozone monitor interferences, J. Air Waste Manage., 60, 1353–1364, 2010.
Long, R. W., Whitehill, A., Habel, A., Urbanski, S., Halliday, H., Colón, M., Kaushik, S., and Landis, M. S.: Comparison of ozone measurement methods in biomass burning smoke: an evaluation under field and laboratory conditions, Atmos. Meas. Tech., 14, 1783–1800, https://doi.org/10.5194/amt-14-1783-2021, 2021.

**Response:** Thank you for your comment. This is a valid point. In response, we have added content to both the main text and supplement and include a relevant figure (Figure S2 within the Supplement).

Fortunately, our ECHAMP peroxy radical sensor provides a separate measurement of Ox (Ox = $O_3$ + $NO_2$) which we can compare to the 2B $O_3$ measurements. In the ECHAMP inlet, ambient air is mixed with excess NO which reacts with $O_3$ to form $NO_2$ which is later quantified by an Aerodyne CAPS sensor based on absorption of light at 450 nm. Unlike the 254 nm absorption bandpass used by the UV absorption method, few compounds absorb in this blue region of the spectrum and so these positive interferences are expected to be minimal (Kebabian et al., 2008). In new SI Figure S2, we share a time series of both the 2B O3 and one of the CAPS Ox results. For a more direct comparison, we added $NO_2$ (measured separately by TILDAS) to our 2B $O_3$ measurement to generate concentrations of Ox. The 17 August period shown includes a non-smoke period followed by a smoke period that starts at about 15:30 local time. The CAPS instrument went offline for diagnostic purposes at 19:30. Organic PM mass concentrations increased from ~10 ug m⁻³ to 30 ug m⁻³ upon smoke influence. There is good agreement between the 2B derived Ox and CAPS Ox during both the smoke and non-smoke periods for both this period shown and for the entire campaign, indicating that interferences in the 2B Tech $O_3$ instrument were minimal. This is not surprising as the smoke plumes we sampled were more dilute than those considered by Long et al. (2021). Here, the smoke plumes led to CO enhancements of near 100 ppb, whereas the smoke plumes studied by Long et al. (2021) had CO concentrations of several parts per million.

**Supplement (new section has been added: *S2 Supplementary Measurements*)**

**S2 Supplementary Measurements**

Photometric $O_3$ monitors can suffer from interferences from VOCs and PM that absorb or scatter 254 nm UV radiation (Huntzicker and Johnson, 1979). Recent measurements in concentrated BB plumes ([CO] > 1 ppmv) have shown that this interference can be large for certain types of photometric $O_3$ measurements (Long et al., 2021). The potential for our $O_3$ measurements, collected using a 2B-Tech Model 205 UV absorption monitor, to be impacted is low given that we sampled dilute BB plumes in this study (enhancements of PM < 30 µg m$^{-3}$ and [CO] < 0.1 ppmv). Furthermore, a comparison of the 2B $O_3$ measurements to separate Ox measurements made by our novel ECHAMP $XO_2$ sensor demonstrate that interferences are negligible. ECHAMP measures Ox by mixing ambient $O_3$ with excess NO to form $NO_2$ which is later quantified by CAPS $NO_2$ instruments. The CAPS Ox results are based on the absorption of light at 450 nm and are expected to have minimal interferences as few compounds absorb in this region (Kebabian et al., 2008). We show a time series of 2B-Tech $O_3$ and ECHAMP Ox measurements in Fig. S2. The Ox measurements only include up to 2 ppbv $NO_2$, and a separate Ox time series derived from adding the 2B-Tech $O_3$ with TILDAS $NO_2$ data is shown for a more direct comparison to the ECHAMP CAPS Ox data. The 17 August period includes nearly 15 hours of background air and a nearly 3-hour smoke influenced period that started at 15:27 MDT and included enhancements of ~100 ppbv CO and ~20 µg m$^{-3}$ organic PM. The 2B $O_3$ data during both the background and smoke periods, as well as for the entire campaign, agree with the CAPS Ox acquired from ECHAMP.

**New Figure included in Supplement:**

[Figure]

**Figure S2.** Time Series of 2B-Tech $O_3$, 2B-Tech derived Ox, and CAPS Ox data acquired from ECHAMP $XO_2$ sensor.

---

## Author Response (AR2)

**Point-by-point Response to Reviewer's Comments**

We appreciate the reviewers for taking time to carefully review the manuscript resubmission.

Below are point-by-point responses to Referee #1 and Referee #3.

**Report #1 by Anonymous Referee #1**

The reviewer thanks the authors for carefully addressing the comments. Also, the reviewer would like to clarify that there is no intention to undermine the importance of the XO2 measurements. The authors' efforts to make this important and challenging measurement are well appreciated.

Since the initial submission of this manuscript, the following two studies on radical chemistry in wildfire ozone have been published. Given their relevance, the reviewer recommends them to be cited in this manuscript.

1. Robinson et al., Variability and Time of Day Dependence of Ozone Photochemistry in Western Wildfire Plumes. Environ Sci Technol 55, 10280-10290 (2021).

2. Xu et al., Ozone chemistry in western U.S. wildfire plumes. Science Advances 7, eabl3648 (2021).

**Response:** These recent papers are quite relevant, and so we have cited them.

**Robinson et al.**, was cited in line 104:

BB emissions also include unique VOCs that are typically unaccounted for by chemical mechanisms employed by models. For instance, the importance of furanoids for model predictions of secondary pollution formation has only recently been studied (Müller et al., 2016;Coggon et al., 2019;Decker et al., 2019;Salvador et al., 2021;Robinson et al., 2021).

and in line 296:

Our calculated values for P(ROx) are limited by the lack of measurements for HONO, which is the dominant HOx source in freshly emitted BB smoke (Peng et al., 2020;Robinson et al., 2021).

**Xu et al.**, was cited in line 97:

Ozone formation in young BB plumes is, in almost all cases, initially NOx-saturated (VOC-limited) but transitions to being NOx-limited as the NOx is photochemically processed to nitric acid and organic nitrates (Xu et al., 2021;Alvarado et al., 2015;Müller et al., 2016;Folkins et al., 1997).

and in line 375:

Based on literature trends where ΔO3/ΔCO values increase with smoke age until an eventual plateau (Jaffe and Wigder, 2012;Baker et al., 2016;Xu et al., 2021), …

**Report #3 by Anonymous Referee #3**

The authors did a good job of responding to most of my comments. However, I still find my second comment regarding the apparent correlation between O3 and HCN in Fig. 4 to be unresolved, and I have a few follow up thoughts about my first comment on the role of the non-fire-related diurnal O3 cycle.

Regarding my first comment on the effect of the normal daily O3 cycle on the data in Fig. S8, I just want to acknowledge that I think the authors' strategy of omitting the 22 and 23 August data from the Fig. S8 analysis is certainly a great improvement, as those two days were most clearly impacted by normal daily O3 production. The remaining trends from the 16, 17, and 24 Aug data are more likely to be driven by smoke-related influence. I am less confident than the authors state they are that it is actually smoke related, but the text has been updated with language regarding the daily O3 cycle so readers can form their own opinions. Hopefully future analyses will make an attempt to use modeling to differentiate the natural O3 production from the smoke-related O3 production (which would be too much additional effort for this manuscript). Thank you for considering my comment.

Regarding my second comment on the correlation between O3 and HCN, I find the authors' response to be missing the point of my comment. Perhaps my comment was not clear enough, so I will try again. I understand that my comment about the impact of a smoke plume arriving at nighttime is irrelevant to a plot showing only daytime data, but the point I was trying to make was that the trends in Fig 4 will be driven by the time of day that the smoke arrives. The main reason that Fig. 4 has a positive slope is because the smoke plumes arrived predominantly in the afternoons for the days shown. For instance, the data for 22 and 23 aug show positive slopes for each day because the data in the morning starting at 9:00 MDT had low O3 and low smoke and HCN. As the day progressed to afternoon, HCN increased due to smoke arriving and O3 increased due to a combination of the natural daily cycle and smoke influence. The O3 increase is likely dominated by the natural daily cycle, but by showing Fig. 4 you are implying that increased O3 is due to smoke influence. I don't believe this is the correct plot to show to draw that conclusion.

The data on 24 Aug in Fig. 4 is another good example. During the morning to midday hours, smoke arrived coincident with increasing daily O3, giving a positive slope to that portion of the data. However in this case, the smoke receded during the afternoon hours so HCN dropped dramatically but O3 remained roughly constant or perhaps dropped by only a few ppb (which may have been part of the natural evening decrease in O3). This portion of the data gives a slope near zero. If the smoke influence was the reason for increased O3, you would expect O3 to drop along with HCN as the smoke left the area. The data on 18 and 21 Aug give slopes that are nearly vertical, because O3 increased from the normal daily cycle while the smoke influence remained minimal.

Going back to my hypothetical example of a smoke plume arriving during nighttime, I will amend that to say it arrives instead at 9:00 MDT. HCN will start elevated, but O3 will not be elevated because daytime photochemistry has only just begun as the sun rose. Say the plume moved out of the area by midday, and HCN has decreased while O3 has increased to a daily maximum. The slope of the data for this day would be included in Fig 4 and would be negative.

Because of this reasoning, I believe the apparent positive correlation between O3 and HCN is a coincidence because the smoke tended to arrive in the mid afternoon at this site during this measurement period (and probably at most sites, because fires tend to burn most in the afternoon and into evening). By plotting all of the data from different days together in the same plot, the details I describe above are lost and the overall trend is driven mainly by the fact that 23 Aug happened to have both the highest O3 and the highest HCN during the afternoon hours. Was there more O3 because there was more smoke, or was it a particularly hot day that caused more natural O3 production as well as a larger fire with smoke? That is a question that cannot be answered with the analysis in Fig. 4. But by showing Fig. 4 and stating at line 419 (in the tracked changes doc) that "For the entire campaign, there is a positive correlation between daytime O3 (and Ox) concentrations and smoke tracer HCN", you are implying that smoke causes increased O3, or at least a reader is very likely to draw that improper conclusion from the plot when presented this way. It is definitely possible that smoke does lead to a relatively small amount of additional O3, but this analysis is not separating the natural photochemical cycle from the smoke-related increase, and thus Fig 4 and the paragraph at line 419 are misleading to the reader.

To address this comment, I have several suggestions. Figure 4 and the associated paragraph could probably just be removed, since they are not really critical to any of the other analyses in this paper. The authors could perform additional modeling to try to model O3 with and without the influence of smoke for the days of this study, but that is likely to be an entire manuscript on its own and too much additional work for this manuscript. Perhaps a middle ground would be to somehow alter Fig. 4 to try to remove/limit the impacts from the natural daily O3 cycle, as they did in response to my other main comment in the first review. E.g., they could restrict the plot to only show data for the hour or several hours during midafternoon when O3 typically peaks. This may show that peak daily O3 correlates positively with HCN, in a way that is more likely to be a true influence from smoke than the way Fig. 4 is currently shown. This method still could be misleading, because as I mentioned in the previous paragraph both O3 and increased smoke could just be the result of a hotter day or the specific photochemical conditions. So I would suggest the authors do some basic checks to see if that could be the case. I suggest they at least explicitly advise the reader in the text that there are external reasons why O3 and HCN could increase together due to external factors, rather than giving the reader the impression that the data is proving that smoke is the dominant cause for increased O3.

I'll thank the readers in advance for addressing this comment again.

**Response:**

We have considered the reviewer's concerns regarding the $O_3$-HCN correlation, and in response, we have updated Fig. 4 and the relevant text.

The updated Fig. 4 now includes further time restricted $O_3$-HCN comparisons (Fig. 4b and Fig. 4c). This follows the 'middle ground' approach suggested by the reviewer and was prepared for 2-hr periods that often had smoke influence. We have opted to keep the original figure (Fig. 4a using all daytime $O_3$ and HCN data). We still consider the overall $O_3$-HCN correlation to be apparent, though we make sure to include several caveats regarding this correlation. Following the reviewer's concerns, we state that the overall positive $O_3$-HCN correlation is partially coincidental due to smoke presence (high HCN) occurring at times that $O_3$ would be high in the absence of smoke. The updated Fig. 4 and its relevant text (new text is shown highlighted) are included below.

**Revised Figure 4:**

[Figure]

**Figure 1** Correlation between $O_3$ and smoke tracer HCN for all observations between 9:00 and 22:00 MDT (panel a), 14:00 and 16:00 MDT (panel b), and 16:00 and 18:00 MDT (panel c). Data points are colored by date collected.

**Revised Text (line 369):**

For the entire campaign, there is a positive correlation between daytime $O_3$ (and Ox) concentrations and smoke tracer HCN (Fig. 4a) with the highest values for both observed on 23 August. Most periods of elevated HCN occurred during the times of day when [$O_3$] was usually high even in the absence of smoke (afternoon or early evening), so the overall positive correlation between $O_3$ and HCN may be partially coincidental. The positive correlation remains, however, when the analysis is restricted to 2-hour periods of afternoon and early evening data to limit the time-of-day dependence (Fig. 4b and Fig. 4c). These more specific $O_3$-HCN comparisons remain impacted by day-to-day variability in $O_3$ from changes in background $O_3$ values, meteorology, and BB emissions of HCN and $O_3$ precursors. Smoke age also plays a role in this correlation plot. Based on literature trends where $\Delta O_3/\Delta CO$ values increase with smoke age until an eventual plateau (Jaffe and Wigder, 2012;Baker et al., 2016;Xu et al., 2021), young smoke plumes are likely to have smaller $O_3$ enhancements relative to smoke tracers like HCN compared to aged plumes. Clusters of data points at HCN concentrations below 0.75 ppbv are observed for the 18 August and 24 August data sets. For 18 August, there was no distinct BB influenced period and a minimal range in [HCN]. This led to the cluster of 18 August data points with [$O_3$] near 60 ppbv. The 24 August data cluster near 50 pbbv [$O_3$] captures the stable period after [$O_3$] is depleted by ~7 ppbv upon smoke departure followed by a slow build in concentration.A similar figure with $O_3$ plotted against CO but for times specific to the arrival or departure of smoke is shown in the SI (Fig. S8).